# How inhibitory and excitatory inputs gate output of the inferior olive

Sebastián Loyola[1,2,3][*][†], Tycho M Hoogland[1,2][*][†], Hugo Hoedemaker[2], Vincenzo Romano[1], Mario Negrello[1][*][†], Chris I De Zeeuw[1,2][*][†]

[1]Department of Neuroscience, Erasmus MC, Rotterdam, Netherlands; [2]Netherlands Institute for Neuroscience, Royal Academy of Arts & Sciences, Amsterdam, Netherlands; [3]Faculty of Health Sciences, Universidad Católica Silva Henríquez, Santiago, Chile

*For correspondence:
sebastian.loyola.arroyo@gmail.com (SL);
tmhoogland@gmail.com (TMH);
m.negrello@erasmusmc.nl (MN);
c.dezeeuw@erasmusmc.nl (CIDZ)

[†]These authors contributed equally to this work

Competing interest: The authors declare that no competing interests exist.

**Abstract** The inferior olive provides the climbing fibers to Purkinje cells in the cerebellar cortex, where they elicit all-or-none complex spikes and control major forms of plasticity. Given their important role in both short-term and long-term coordination of cerebellum-dependent behaviors, it is paramount to understand the factors that determine the output of olivary neurons. Here, we use mouse models to investigate how the inhibitory and excitatory inputs to the olivary neurons interact with each other, generating spiking patterns of olivary neurons that align with their intrinsic oscillations. Using dual color optogenetic stimulation and whole-cell recordings, we demonstrate how intervals between the inhibitory input from the cerebellar nuclei and excitatory input from the mesodiencephalic junction affect phase and gain of the olivary output at both the sub- and suprathreshold level. When the excitatory input is activated shortly (~50 ms) after the inhibitory input, the phase of the intrinsic oscillations becomes remarkably unstable and the excitatory input can hardly generate any olivary spike. Instead, when the excitatory input is activated one cycle (~150 ms) after the inhibitory input, the excitatory input can optimally drive olivary spiking, riding on top of the first cycle of the subthreshold oscillations that have been powerfully reset by the preceding inhibitory input. Simulations of a large-scale network model of the inferior olive highlight to what extent the synaptic interactions penetrate in the neuropil, generating quasi-oscillatory spiking patterns in large parts of the olivary subnuclei, the size of which also depends on the relative timing of the inhibitory and excitatory inputs.

## Editor's evaluation

Inferior olivary neurons drive complex spiking activity in Purkinje neurons of the cerebellar cortex, ultimately playing critical roles in controlling motor coordination and plasticity. Using transgenic mice or optogenetic techniques to independently control a major excitatory and inhibitory pathway to the inferior olive, the authors show solid evidence that the probability and phase of olivary neuron output depend critically on the relative timing of excitation and inhibitory inputs. Network models predict that appropriately timed excitatory and inhibitory input patterns efficiently synchronize larger clusters of inferior olivary neurons, raising the possibility that input timing can gate the output of the motor commands. These valuable findings have the potential to impact the field's understanding of sensorimotor processing.

## Introduction

The inferior olive (IO) forms an integral part of the cerebellar system in that it provides all the climbing fibers innervating Purkinje cells in the cerebellar cortex (**Ito, 1984**). Activation of the climbing fiber

terminals elicits both short-term and long-term effects (*De Zeeuw et al., 2011*). In the short-term, it evokes powerful, all-or-none, depolarizations in Purkinje cells also known as complex spikes (*Thach, 1967*); increases in complex spike activity within cerebellar microzones can facilitate reflex-like movements, such as fast reactive limb movements following a perturbation (*De Gruijl et al., 2014a*; *Hoogland et al., 2015*), but when occurring in more widespread ensembles they also mediate more demanding forms of motor behavior, such as skilled sequences of contractions of tongue muscles upon presentation of specific sensory cues (*Welsh et al., 1995*; *Bina et al., 2022*; *Romano et al., 2022*). In the more long term, modulation of climbing fiber activity facilitates the induction of various forms of pre- and postsynaptic plasticity at most, if not all, parallel fiber synapses onto interneurons and Purkinje cells in the molecular layer of the cerebellar cortex (*Gao et al., 2012*). Due to the fact that these different subcellular effects occur in a synergistic fashion in that they ultimately converge into either a downbound or upbound impact on simple spike activity in the Purkinje cells (*De Zeeuw, 2021*), climbing fibers are also prominently implicated in cerebellar learning (*Ito and Kano, 1982*; *Raymond and Lisberger, 1998*; *Safo and Regehr, 2008*; *ten Brinke et al., 2015* and *ten Brinke et al., 2019*; *Gutierrez-Castellanos et al., 2017*; *Boele et al., 2018*; *Bina et al., 2022*; *Romano et al., 2018*). For both the short-term and long-term effects in the cerebellar cortex the precise timing of the climbing fiber activation is crucial. For example, when the synchronization of complex spikes within microzones is enhanced, the latency of subsequent reflex movements will be shortened (*Kistler et al., 2002*; *Van Der Giessen et al., 2008*; *De Gruijl et al., 2014a*). Or likewise, depending on the timing of the activity of the climbing fiber input with respect to that of the parallel fiber input, depression or potentiation plasticity mechanisms in Purkinje cells will prevail (*Wang et al., 2000*; *De Zeeuw, 2021*).

Given the paramount impact of precise timing of climbing fiber activation, it is important to identify the factors that contribute to this process. Classical work by Llinas and Yarom has revealed that olivary neurons are endowed with dendro-dendritic gap junctions as well as ionic conductances that allow them to oscillate in synchrony (*Llinas et al., 1974*; *Sotelo et al., 1974*; *Llinás and Yarom, 1981*; *Llinás and Yarom, 1986*; *Lampl and Yarom, 1993*; *Lampl and Yarom, 1997*; *Leznik and Llinás, 2005*; *Choi et al., 2010*). More specifically, olivary neurons can show different types of subthreshold activity (*Khosrovani et al., 2007*; *Turecek et al., 2014*), which can partly be explained at the cellular level by differential expression of Cav 3.1 channels (T-type $Ca^{+2}$) (*Urbano et al., 2006*; *Bazzigaluppi and de Jeu, 2016*). Whole-cell recordings in vivo, systems' extracellular recordings, as well as modeling studies indicate that olivary subthreshold oscillations (STOs) are likely to occur at preferred frequencies below 10 Hz for a limited number of cycles and that the complex spikes ride on this rhythm (*Sasaki et al., 1989*; *Bal and McCormick, 1997*; *Blenkinsop and Lang, 2006*; *Chorev et al., 2007*; *Choi et al., 2010*; *Lefler et al., 2014*; *Turecek et al., 2014*; *Turecek et al., 2016*; *Lang et al., 2017*; *Khosrovani et al., 2007*; *Negrello et al., 2019*). Question remains though, how and to what extent the afferents to the olivary neurons modify the intrinsic pattern of STOs and thereby that of their spiking activity (*Placantonakis et al., 2006*; *Best and Regehr, 2009*; *Bazzigaluppi et al., 2012a*; *Bazzigaluppi et al., 2012b*; *Garden et al., 2017*; *Garden et al., 2018*). The vast majority of the axon terminals of the afferents target the dendritic spines located in the olivary glomeruli (*Sotelo et al., 1974*; *de Zeeuw et al., 1989*). Many of these spines are electronically coupled by gap junctions and most, if not all, of them receive both an inhibitory and excitatory input (*de Zeeuw et al., 1989*; *De Zeeuw et al., 1990*). Based upon a theoretical model of excitable spines in such configuration (*Segev and Parnas, 1983*; *Segev and Rall, 1988*), it has been advocated that spike generation of olivary neurons may be highly sensitive to the temporal interval between activation of the inhibitory and excitatory inputs (*De Zeeuw, 1990*; *De Zeeuw et al., 1998*).

Using whole-cell patch-clamp recordings of olivary neurons in combination with dual color optogenetics, we set out to test this hypothesis in mice by investigating the temporal response properties of neurons in the principal olive (PO) and rostral medial accessory olive (MAO), both of which receive their monosynaptic, inhibitory and excitatory inputs from the cerebellar nuclei (CN) and mesodiencephalic junction (MDJ), respectively (*de Zeeuw et al., 1989*; *De Zeeuw et al., 1990*; *Ruigrok and Voogd, 1990*; *Apps and Hawkes, 2009*; *Lefler et al., 2014*). Our data indicate that optimal and stable control of olivary spiking behavior indeed strongly depends on the temporal interval between cerebellar inhibition and mesodiencephalic excitation, allowing bidirectional control of firing probability. The sensitivity to this interval is so high that stimulation of the excitatory input at the appropriate moment following inhibition triggers more olivary spikes than stimulation of the excitatory input

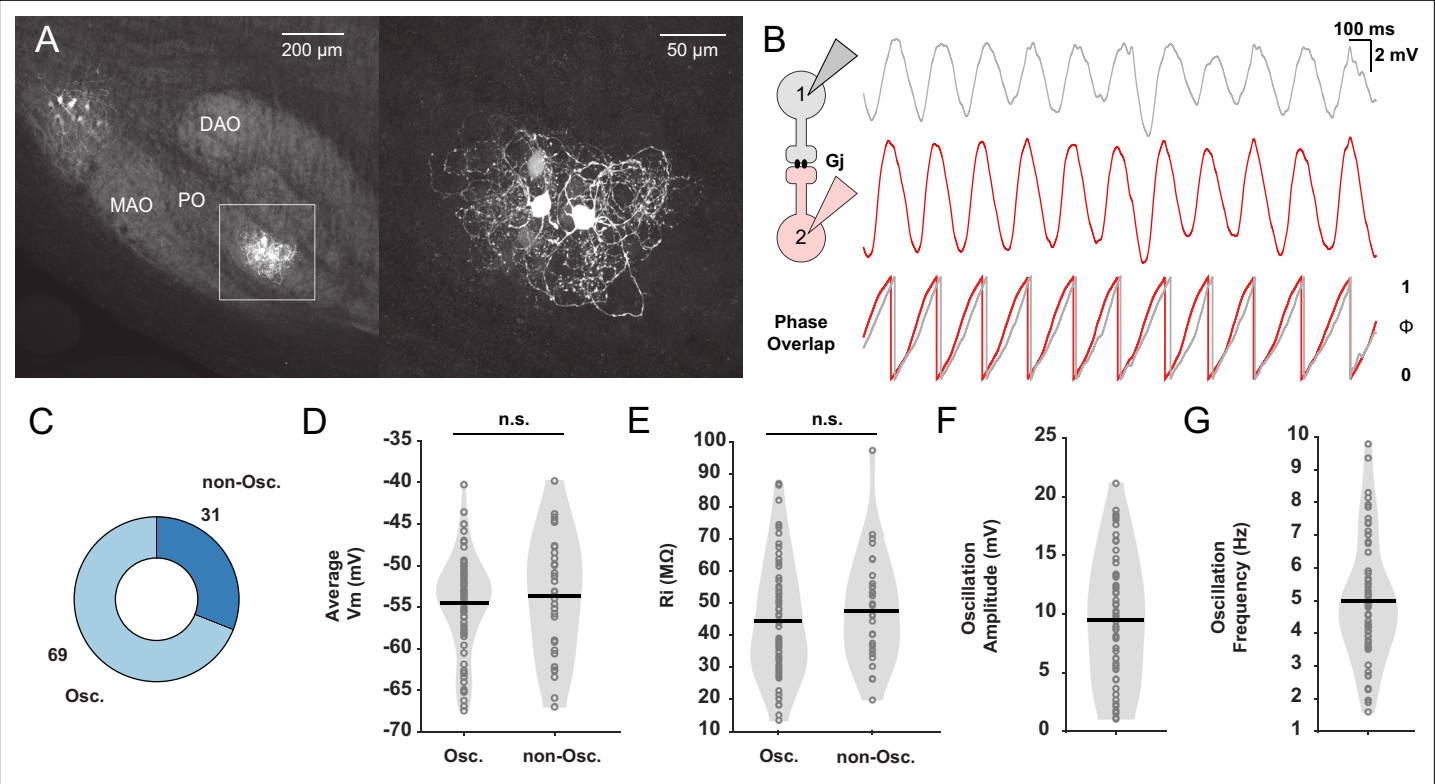

**Figure 1.** In vitro whole-cell patch-clamp recordings of oscillating and non-oscillating neurons in the inferior olive (IO). (**A**) Dye coupling between electrically coupled neurons in the IO was visualized by performing immunostaining on neurons dialyzed with biocytin. Note that the dendrites of the cluster of the labeled olivary neurons are in this case exclusively located in the principal olive (PO) and do not traverse into the medial accessory olive (MAO) or dorsal accessory olive (DAO). Right panel corresponds to inset on the left. (**B**) Left: Schematic illustrating dual whole-cell patch-clamp recordings of two IO neurons coupled by gap junctions (Gj). Right: Example traces of two IO neurons displaying synchronized subthreshold oscillations (STOs) and the phase of these oscillations with respect to each other underneath. (**C**) Pie plot showing the proportion (in %) of IO neurons that display STOs (Osc.) or not (non-Osc.). (**D**) Average membrane potential (Vm) in millivolt (mV) of oscillating and non-oscillating neurons. (**E**) Input resistance (Rin) in mega-ohm (MΩ), of oscillating and non-oscillating neurons. (**F and G**) Amplitude (in mV) and frequency (in Hz) of STOs of oscillating IO neurons, respectively. Each open gray circle in panels D to G represents one cell; black lines indicate mean values; and n.s. indicates not significant.

alone. The in vitro experiments confirmed by simulations running on a realistic model of the mouse IO indicate that this high sensitivity results at least in part from the fact that inhibition induces a strong phase reset of the STOs on top of which the excitatory input can drive the spiking output. Moreover, the simulations of coupled olivary neurons also suggest that the size of the olivary network that is engaged in generating a particular firing pattern also depends on the timing and overlap between both inputs. In short, the relative timing of the inhibitory and excitatory inputs with respect to each other determines the timing of the spiking output of the olivary neurons, gating the impact of their climbing fibers in the cerebellar cortex.

## Results
### Cell physiological properties
Prior to investigating how IO neurons integrate synaptic inputs, we characterized their physiological properties in sagittal sections of the brainstem (*Figure 1A–G*). Recordings were obtained from single cells or pairs of coupled cells in the PO and rostral part of the MAO using whole-cell patch-clamp recordings (*Figure 1A and B*). To estimate the number of neurons to which a particular neuron is electrotonically coupled by gap junctions, we included biocytin (0.5%) in our patch pipette, a molecule sufficiently small to cross gap junctions formed by connexin 36 (*Van Der Giessen et al., 2006*). Using this approach in slices of 300 µm thickness, we obtained an average of 5.1±6.9 dye-coupled neurons

per patched cell (range: 0–27; n=20, N=17), which is in line with previous in vitro studies (*De Zeeuw et al., 2003*; *Placantonakis et al., 2006*; *Turecek et al., 2014*).

Across all recordings 69% of the IO neurons (69/100 cells; N=43) displayed STOs during spontaneous activity (*Figure 1C*). The oscillating and non-oscillating cell groups did not differ in their passive membrane properties (*Figure 1D and E*). Neither the membrane potential (oscillating cells: an average of –54.52±5.69 mV ranging from –67.29 to –40.24 mV; non-oscillating cells: an average of 53.62±7.14 mV ranging from –66.86 to –39.8 mV) nor the input resistance (oscillating cells: an average of 44.49±17.83 MΩ ranging from 13.82 to 87.09 MΩ; non-oscillating cells: an average of 47.64±16.64 MΩ ranging from 20.2 to 97.46 MΩ) differed significantly among the two cell types (p=0.49 and p=0.40, respectively; unpaired t-test). Oscillations of cells with STOs had a mean amplitude of 9.52±5.4 mV (range: 1.13–21.14 mV) and a mean frequency of 4.97±1.85 Hz (ranging from 1.63 to 9.77 Hz) (*Figure 1F and G*).

## CN input and subthreshold activity of IO neurons

IO responses to activation of the inhibitory synapses from the CN consist of a GABAAR-mediated hyperpolarization (Vhyp) followed by a rebound depolarization (Vreb), reliant on low voltage-gated T-type calcium channels and HCN channels (*Choi et al., 2010*; *Bal and McCormick, 1997*). To selectively stimulate the GABAergic afferents from the CN in the IO with optogenetics, we used either crossbreeds of Gad2-IRES-Cre and Ai32(RCL-ChR2(H134R)/EYFP) mouse lines (*Fink et al., 2014*; *Berndt et al., 2011*) or C57/BL6 mice in which the CN was injected 4 weeks prior to the experiments with a virus containing ChrimsonR-tdTomato opsin-fusion protein (*Klapoetke et al., 2014*; *Figure 2A and B*). With either approach optogenetic stimulation of the afferents in vitro readily evoked biphasic responses consisting of the Vhyp and Vreb components (*Figure 2C*). Depending on the depolarized state, a synaptically induced rebound depolarization could reach threshold and elicit an olivary spike. Half-widths of Vhyp and Vreb of IO cells found in the mice virally transduced with ChrimsonR did not differ significantly from those in the transgenic mice expressing ChR2 (for Vhyp and Vreb p=0.29 and p=0.72, respectively; unpaired t-test) (*Figure 2—figure supplement 1A*), but the onset latencies of the synaptic responses following optogenetic stimulation were shorter with the viral approach (p<0.0001; Mann-Whitney U test) (*Figure 2—figure supplement 1A*). Application of the GABAA receptor antagonist picrotoxin completely blocked Vhyp and Vreb (n=5 and N=4; *Figure 2—figure supplement 1*), underscoring the GABAergic nature of the response.

Optogenetic stimulation (20 ms light pulses, 0.4–3.85 mW/mm$^2$) of CN afferents in the IO in vitro induced transient oscillations in 38.2% of non-oscillating cells (12 out of 31 cells; N=25), which is in line with previous data obtained in vivo (*Khosrovani et al., 2007*; *Bazzigaluppi and de Jeu, 2016*). Such oscillations, which lasted up to 2.88 s (average duration of 1.8±0.65 s; n=5, N=4), could only be induced if the hyperpolarizing component of the synaptic potential (Vhyp) had a magnitude of at least 1.14 mV (*Figure 2D*). On average, the amplitude of the first cycle following stimulation (*Figure 2D*, lower panel) scaled well with the amplitude of Vhyp (r$^2$=0.95; n=5, N=4) (*Figure 2E*, first panel) and was modulated by membrane voltage (*Figure 2—figure supplement 2A and B*). Subsequent cycles showed lower correlation levels (for second and third cycles, r$^2$=0.53 and 0.49, respectively; n=5, N=4). Likewise, the latencies to the peak of the rebound (*Figure 2D*, upper panel) as well as the slope of the rebound (*Figure 2C*) could to some extent be correlated to Vhyp, even though different categories became apparent (*Figure 2E*, second and third panels). Possibly, these categories reflect the differential speeds in desensitizing responses that are mediated by GABAA receptors of different subunit composition (*Devor et al., 2001*). Together, our findings in IO cells without STOs suggest that CN afferent stimulation induces hyperpolarizing and rebound responses, and that the magnitude of these responses determines the amplitude, latency, and duration of the induced transient oscillations.

Next, we investigated the impact of CN input on the phase changes (Δφ) of IO neurons with spontaneous STOs (*Figure 2F*). When stimulating CN afferents at various incident phases (φ) of the STO cycle, we extracted two parameters, the slope and Δφ-intercept, from the phase response curve (PRC) (*Figure 2G*). The slope provides a measure of how sensitive phase changes are to the incident phase at which the afferent input arrives, while the Δφ-intercept provides insight into whether stimulus-induced phase changes consist exclusively of phase delays (negative intercepts) or delays and advances (positive intercepts). In the subthreshold regime, CN afferent stimulation acted as a strongly resetting stimulus for oscillating IO neurons and consistently introduced a phase delay for each phase

**Figure 2.** Responses of oscillating and non-oscillating neurons in the inferior olive (IO) to stimulation of GABAergic afferents from the cerebellar nuclei (CN). (**A**) Either we expressed ChrimsonR-tdTomato in the CN of wild type mice or we use GAD2Cre/Chr2- H134R-EYFP transgenic mice so as to be able to specifically stimulate the GABAergic input to the IO with optogenetics. (**B**) Confocal images of a sagittal section of the brainstem comprising the IO. Note that the GABAergic fibers from the CN project to all three olivary subnuclei, including the principal olive (PO), medial accessory olive (MAO), and dorsal accessory olive (DAO). Insets on the left correspond to higher magnifications on the right. (**C**) Stimulation of CN fibers typically elicits a biphasic response in olivary neurons; this response starts with a hyperpolarizing component (Vhyp) and is followed by a rebound component (Vreb). Arrows denote how the amplitude was measured for each component. (**D**) Whole-cell patch-clamp recording of an IO neuron that did not oscillate during baseline (i.e., a non-oscillating cell). Note that the cell starts to oscillate upon CN stimulation (pink arrow-head) and that the amplitude of the oscillation (presented as Vhyp) depends on the strength of stimulation (top, weak; bottom, strong). Dark pink traces indicate average responses; light pink traces indicate individual responses. (**E**) Amplitude of the first cycle of the oscillation (first panel), latency to its peak from the onset of the stimulus (referred to as latency to rebound peak, second panel), and slope of this first cycle (referred to as rebound slope, third panel) following CN stimulation as a function of the amplitude of the hyperpolarizing component (Vhyp). Open pink circles represent individual trials obtained from five cells. Schemes showing how these parameters were calculated are shown in panel D (first cycle amplitude and latency to rebound peak) and panel C (rebound slope was calculated from peak of Vhyp to the peak of Vreb). (**F**) Superimposed traces of whole-cell patch-clamp recordings of an IO neuron that oscillates during baseline (i.e., an oscillating cell) (top panel). CN stimulation (pink arrow-head) enhances the amplitude of the oscillation and evokes a prominent resetting of the phase (see phase plot in bottom panel). (**G**) Phase shifts of subthreshold responses (y-axis) evoked by CN stimulation given at particular phases of the oscillations during baseline activity (x-axis). Open pink circles represent individual trials from five cells, together forming the phase plot curve. (**H**) Histogram showing the distribution of phase lags calculated across trials for three cycles before (pre-stimulus) and three cycles after (post-stimulus) CN stimulation. (**I**) Changes in amplitude of the four cycles of subthreshold oscillations (STOs) before and after CN stimulation. Shaded areas represent SD.

The online version of this article includes the following figure supplement(s) for figure 2:

**Figure supplement 1.** Picrotoxin blocks the cerebellar nuclei (CN) synaptic response.

**Figure supplement 2.** Voltage dependence of the rebound depolarization (Vreb) amplitude of cerebellar nuclei (CN) and mesodiencephalic junction (MDJ) synaptic responses.

of the cycle (slope = –0.52 ± 0.01, ΔΦ-intercept = 0.07 ± 0.01, $r^2$=0.86; n=5, N=3) (**Figure 2G**). To estimate the impact on the regularity of phase resetting across trials, we calculated the intertrial phase jitter across three oscillation cycles before and after inhibitory stimulation and compared the standard deviations of the pre- and post-stimulus phase lags (from now on referred to as SDpre-stim and SDpost-stim, respectively). Across trials, intertrial phase jitter was significantly reduced after stimulation (SDpre-stim=59.51 ms, SDpost-stim=20.35 ms, n=15, N=14; p<0.0001, F test), indicating that the stimulus induced nearly identical post-stimulus phase responses across trials (**Figure 2F and H**). To determine to what extent subthreshold inhibitory stimulation affected STO amplitude, we next quantified the increase in STO amplitude of four post-stimulus cycles relative to a pre-stimulus baseline (set as the average of the amplitude of pre-stimulus cycles). Inhibitory synaptic responses could significantly boost STO amplitude (ΔSTO amplitude of first cycle 4.8±3.7 mV, p<0.0001; second cycle 1.0±1.7 mV, p<0.0001; third cycle 0.57±1.7 mV, p<0.0001; fourth cycle 0.4±1.3 mV, n=24, N=19; p=0.0002, Friedman test) (**Figure 2I**).

## MDJ input and subthreshold activity of IO neurons

Olivary synaptic responses to excitatory input from the MDJ are comprised of a depolarization (Vdep) mediated by AMPA and NMDA receptors (**Mathy et al., 2014**; **Turecek et al., 2014**), followed by a hyperpolarization (Vhyp), mediated by both HCN and SK channels (**Garden et al., 2017**; **Garden et al., 2018**). If sufficiently large, this hyperpolarization also drives a rebound depolarization (Vreb) mediated by HCN and T-type calcium channels, which can reach threshold to initiate a spike. MDJ afferents in the IO were stimulated either electrically or with the use of optogenetics in slices (**Figure 3A**). In case of electrical stimulation (100 µs pulses, 0.1–1.2 mA), a bipolar stimulation electrode was placed in the medial tegmental tract, which contains the fibers that descend from the MDJ to the IO (**Ruigrok and Voogd, 1995**). For optogenetics, AAV-Syn-Chronos (**Klapoetke et al., 2014**) was injected in the MDJ and stimulation (1 ms light pulses, 0.6–1.7 mW/mm²) was done at least 4 weeks post-transduction when the medial and principal olivary subnuclei showed a clear innervation of labeled afferents from the MDJ (**Figure 3B**). With both approaches stimulation of MDJ afferents could readily elicit triphasic synaptic potentials in non-oscillating IO cells (**Figure 3—figure supplement 1A and C**); in virtually all cases these potentials comprised a Vdep, Vhyp, and Vreb component, as highlighted above (**Figure 3C and D**). The synaptic potentials evoked by electrical stimulation had shorter latency onsets than those evoked by light stimulation (p<0.0001, Mann-Whitney U test) (**Figure 3—figure supplement 1A**), but otherwise both types of stimulation evoked similar responses. In general, stimulation of MDJ afferents could only trigger STOs in intrinsically silent IO neurons when Vhyp was larger than ~3 mV. The duration of these transient oscillations (1.3±0.64 s, ranging from 0.44 s to 2.36 s) was on average significantly shorter (p=0.005, unpaired t-test; n=6, N=5) than those occurring upon stimulation of the CN afferents. Moreover, with both electrical and optogenetic stimulation, the amplitude of Vdep scaled linearly with stimulus strength, whereas Vhyp and Vreb scaled with stimulation strength in a slightly less linear manner (**Figure 3—figure supplement 1B–D**). Likewise, the half-widths of the three components were comparable following either type of stimulation (for Vdep p=0.84; for Vhyp p=0.62; and for Vreb p=0.085, unpaired t-test) (**Figure 3—figure supplement 1A**). Like the olivary responses to CN input (**Figure 2E**), the amplitude and slope of the first cycle following MDJ stimulation scaled with the size of the preceding hyperpolarization (first cycle: $r^2$=0.92 and 0.93, respectively; n=8, N=7) and was modulated by membrane voltage (**Figure 2—figure supplement 2C and D**). The latency to peak (rebound peak) was inversely related to the hyperpolarization (**Figure 3E**). However, the amplitudes of the second and third cycles of the elicited oscillations correlated weakly with the amplitude of Vhyp ($r^2$=0.26 and 0.04, respectively), reflecting a higher level of variability in the responses to MDJ stimulation relative to CN stimulation. Accordingly, the transient STOs that occurred in the silent or 'non-oscillating' IO cells upon MDJ stimulation also showed a relatively high level of variability of their phase values (**Figure 3D**). Indeed, it should be noted that CN stimulation had a stronger and more consistent impact on resetting the phase of the oscillations in such cells (**Figure 2D**).

Similarly, in oscillating IO cells, MDJ stimulation induced a weak phase resetting (**Figure 3G**) and significantly reduced the phase jitter (SDpost-stim=44.97 ms versus SDpre-stim=56.47 ms; p<0.0001, F test; n=16, N=14), but this latter reduction was significantly smaller (p<0.0001, F test) than that following CN stimulation (compare **Figure 3F–H** with **Figure 2F–H**). Moreover, MDJ stimulation induced a relatively weak, yet significant, increase of the amplitude of the post-stimulus oscillation

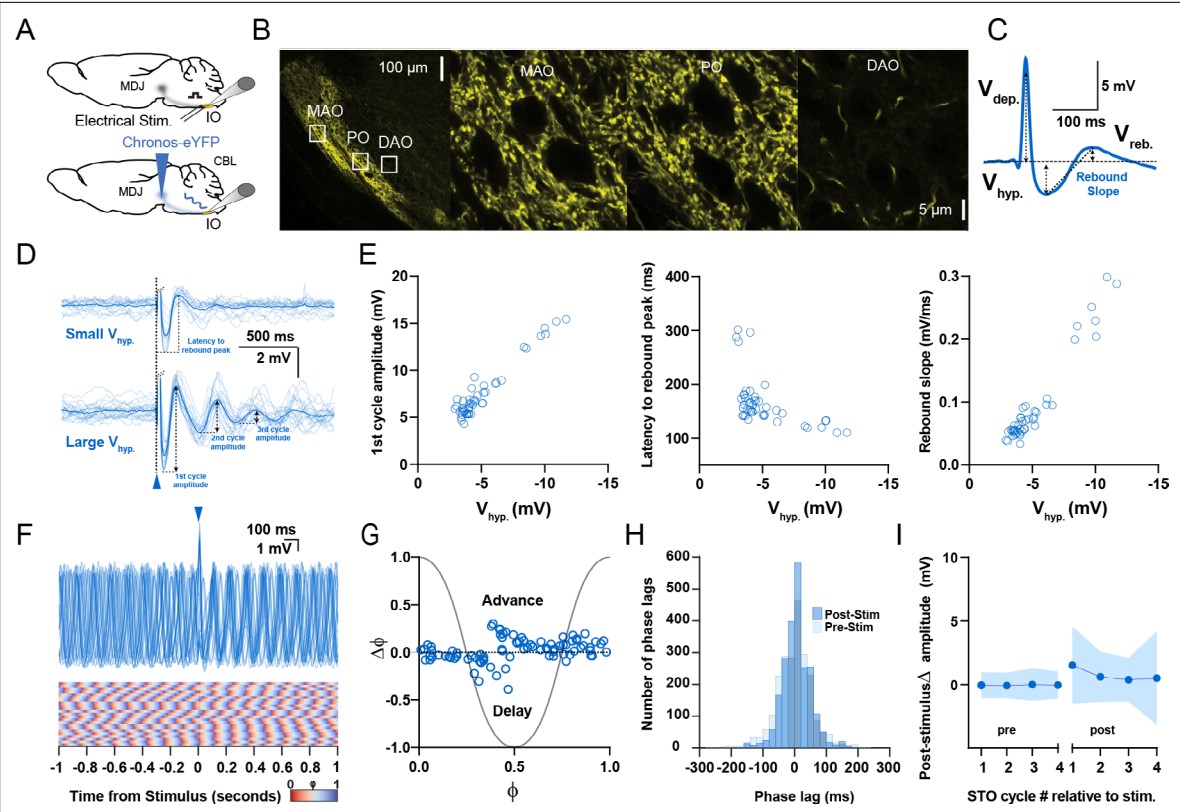

**Figure 3.** Responses of oscillating and non-oscillating neurons in the inferior olive to stimulation of excitatory afferents from the mesodiencephalic junction (MDJ). (**A**) To stimulate neurons in the principal olive (PO) and medial accessory olive (MAO), we used either direct electrical stimulation of the central and medial tegmental tract, respectively (top panel), or optogenetic stimulation following injection of Chronos-eYFP into the MDJ (bottom panel). (**B**) Confocal images of a sagittal section of the brainstem comprising the different olivary subnuclei. Note that the fibers from the MDJ project prominently to the PO and MAO, but not to the dorsal accessory olive (DAO). Insets on the left correspond to higher magnifications on the right. (**C**) Stimulation of MDJ fibers typically elicits a multiphasic response in olivary neurons; the response starts with a depolarizing component (Vdep), followed by a hyperpolarizing component (Vhyp) and a rebound component (Vreb). Arrows denote how the amplitude was measured for each component. (**D**) Whole-cell patch-clamp recordings of a non-oscillating olivary neuron before, during, and after MDJ stimulation, in which synaptic responses with a small Vhyp amplitude elicited virtually no oscillations (top panel), whereas stronger stimulations triggering responses with a larger Vhyp did trigger subthreshold oscillations (STOs) (bottom panel). Dark blue traces indicate average responses; light blue traces indicate individual responses. Part of the depolarizing phase of the synaptic potential (Vdep) has been omitted for clarity in both panels. (**E**) Amplitude of the first cycle of the oscillation (first panel), latency to its peak from the onset of the stimulus (referred to as latency to rebound peak, second panel), and slope of this first cycle (referred to as rebound slope, third panel) following MDJ stimulation as a function of the amplitude of the hyperpolarizing component (Vhyp). Open blue circles represent individual trials obtained from eight number of cells. Schemes showing how these parameters were calculated are shown in panel D (first cycle amplitude and latency to rebound peak) and panel C (rebound slope was calculated from peak of Vhyp to the peak of Vreb). (**F**) Superimposed traces of whole-cell patch-clamp recordings of an individual oscillating cell (top panel). In contrast to CN stimulation (*Figure 2*), MDJ stimulation (blue arrow-head) does not prominently enhance the amplitude of the oscillation. Note that the phase resetting upon MDJ stimulation in the phase plot in the bottom panel is also less clear than that upon CN stimulation. (**G**) Phase shifts of subthreshold responses (y-axis) evoked by MDJ stimulation given at particular phases of the oscillations during baseline activity (x-axis). Open blue circles represent individual trials from eight cells, together forming the phase plot curve. (**H**) Histogram showing the distribution of phase lags calculated across trials for three cycles before (pre-stimulus) and three cycles after (post-stimulus) MDJ stimulation. (**I**) Changes in amplitude of the four cycles of STOs before and after MDJ stimulation. Shaded areas represent SD.

The online version of this article includes the following figure supplement(s) for figure 3:

**Figure supplement 1.** Mesodiencephalic junction (MDJ) evoked synaptic responses in the inferior olive (IO).

cycles in these cells (ΔSTO amplitude of first cycle 1.52±3.0 mV, p<0.0001; second cycle 0.60±2.0 mV, p<0.0001; third cycle 0.37±1.7 mV, p=0.01; fourth cycle 0.5±3.69 mV, p=0.0119; Friedman test, n=26, N=19), (*Figure 3I*). As a result, the increases in oscillation amplitude associated with MDJ afferent stimulation were significantly lower than those following subthreshold CN afferent stimulation (first

cycle p<0.0001; second cycle p<0.0001; third cycle p<0.0001; fourth cycle p<0.0001; Mann-Whitney U test).

Finally, to confirm that MDJ stimulation indeed evoked purely glutamatergic responses, ruling out recruitment of GABAergic fibers, we performed a set of pharmacological control experiments. Vhyp and Vreb were not significantly affected by the GABAA receptor antagonist picrotoxin (amplitude change relative to baseline, Vhyp: 138.1 ± 66.6%, p=0.1493; Vreb: 103.7 ± 36.5%, p=0.7852; n=8, N=8, paired t-test), whereas Vdep slightly increased (122.7 ± 13.92 %, p=0.0024) (*Figure 3—figure supplement 1E and F*). Instead, when AMPA receptors were blocked with CNQX all three components were prominently reduced (Vdep: 22.9 ± 12.18%, p<0.0001; Vhyp: 21.2 ± 19.7%, p<0.0001; Vreb: 10.8 ± 12.6%, p<0.0001; n=8, N=8, paired t-test). Moreover, application of the NMDA receptor antagonist APV (50 µM) caused a further reduction (Vdep: 7.0 ± 9.7%, p=0.001; Vhyp: 13.8 ± 17.4%, p=0.0022; Vreb: 7.2±14.4, p=0.001; n=4, N=4, paired t-test) (*Figure 3—figure supplement 1E and F*). These data establish that stimulation of excitatory MDJ afferents results in synaptic potentials that are predominantly mediated by glutamate receptors (*de Zeeuw et al., 1989*; *Garden et al., 2017*).

## Network model recapitulates STOs' responses to CN and MDJ inputs in isolation

In this work we have extended the IO model from *Negrello et al., 2019*, by scaling the number of cells to capture the murine MAO, comprising over a thousand cells with inhibitory and excitatory arborizations from the CN and MDJ, respectively (*Vrieler et al., 2019*). The IO cells in the model are endowed with gap junctions as well as all the known ion channels that allow them to oscillate and

**Figure 4.** Comparison of subthreshold responses of oscillating olivary neurons obtained with experiments (data) and those found following simulations (model). (**A**) Subthreshold oscillatory responses to stimulation of the cerebellar nuclei (onset indicated by pink arrow-heads). Panels on the left and right show subthreshold activity according to the data and model, respectively; middle panel shows phase plot curves for both data and model. (**B**) As in A, but following stimulation of the mesodiencephalic junction (onset indicated by blue arrow-heads). Note that the model can reproduce both experimental data sets, including the relatively strong and stable phase resetting effect following stimulation from the cerebellar nuclei as compared to that from the mesodiencephalic junction. The experimental data sets in panels A and B correspond to those presented in *Figures 2 and 3*, respectively.

spike (*De Gruijl et al., 2014b*). When applying CN and MDJ stimulation to the oscillating cells of this network model, the PRCs of the subthreshold activities that have been obtained with the experimental approaches (*Figures 2G and 3G*) can be reproduced accurately (*Figure 4*). Indeed, both the strong phase resetting following CN stimulation (*Figure 4A*) and the relatively high variability in phase values following MDJ stimulation (*Figure 4B*) emerge from the model. Even the relatively stable and unstable phase changes at Φ values around 0.5 following CN and MDJ stimulation, respectively, were mimicked in the model (see middle panels of *Figure 4A and B*). These data suggest that the combination of the intrinsic properties characteristic of olivary neurons and the dual inhibitory and excitatory inputs to their dendrites is sufficient to explain our experimental findings.

## Responses of STOs to closely timed CN and MDJ inputs

Given the unique and ubiquitous synaptic arrangement at olivary dendritic spines, most, if not all, of which receive a combined inhibitory and excitatory input (*De Zeeuw et al., 1990*), the temporal interplay of CN and MDJ inputs is bound to have a prominent impact on the input-output relationship of IO neurons. We therefore set out to stimulate CN and MDJ afferents at different intervals with respect to each other and investigated the impact of the stimulus patterns on individual olivary cells. We used two different combinations, both of which allowed us to stimulate the CN and MDJ afferents independently in the same preparation; either we used two different viral vectors with different opsins (AAV-ChrimsonR-tdTomato and AAV-Chronos-eYFP for targeting the CN and MDJ in C57BL/6 mice, respectively) or we combined intrinsic expression of one opsin (for targeting the CN in Gad2-IRES-Cre × Ai32 RCL-ChR2(H134R)/EYFP transgenic mice) with electrical stimulation of the medial tegmental tract (for targeting MDJ fibers in the same mice) (*Figure 5A*). When using the dual opsin approach, slices were obtained at least 4 weeks after the AAV injections to warrant sufficient expression of both opsins in their termination areas of the MAO and PO (*Figure 5B*). We varied the interval between the inhibitory CN and excitatory MDJ inputs, ranging from excitation preceding inhibition by 100 ms up to excitation succeeding inhibition by 200 ms, and we randomly applied these stimulus patterns with respect to the phase of the STOs (*Figure 5C–F*). The PRC slope reflecting phase changes in relation to the phase of the STO during which stimulation occurred did not differ significantly across any of the intervals tested (*Figure 5C and D*), except for the condition where excitation followed inhibition by 150 ms (*Figure 5E and F*); for this interval the slope was significantly steeper (–0.69±0.05; p<0.0001, n=7, N=7, one-way ANOVA) (*Figure 5—figure supplement 1B and C*). Moreover, the ΔΦ-intercept shifted significantly (p<0.0001, one-way ANOVA) from a negative value at an interval of –100 ms (i.e., excitation preceding inhibition; –0.16±0.02; n=6, N=4) to a positive value at an interval of 150 ms (0.35±0.03; n=7, N=7) (*Figure 5C–F, Figure 5—figure supplement 1B and D*). Thus, excitation that follows inhibition by 150 ms can both advance or delay the oscillation phase depending on the phase of the STO at the onset of stimulation, whereas for all the other intervals tested the oscillation phase can only be delayed.

Notably, the largest post-stimulus intertrial phase jitter occurred at the interval where excitation followed inhibition by 50 ms (*Figure 5F*), an interval at which the excitation occurs around the trough of the inhibitory response (*Figure 5D*). Under this condition, the magnitude of the intertrial phase jitter change, which was calculated as the standard deviation post-stimulus minus the standard deviation pre-stimulus (from now on referred to as ΔSDpost-pre), was –28.35 ms (n=34, N=25). Moreover, the intertrial phase jitter change following this stimulus pattern of excitation 50ms after inhibition was comparable to the condition when excitation was provided in isolation (p=0.25, F test) (*Figure 3H*). This indicates that presumably the strong and uniform phase resetting effect caused by inhibition was overruled by excitation at this interval.

Most combinations of inhibition and excitation stimulus patterns increased STO amplitude during the first post-stimulus oscillation cycle (*Figure 5C–F, Figure 5—figure supplement 1G*), presumably reflecting the increased activation of rebound depolarization currents (*Llinás and Yarom, 1981; Choi et al., 2010; Bal and McCormick, 1997*). The changes in post-stimulus STO amplitude were the smallest when excitation was applied 50–70 ms after inhibition (*Figure 5D and F*). Indeed, whereas an interval of +150 ms yielded the highest STO amplitude of 7.25±3.45 mV during the first cycle (n=10, N=8), an interval of +50 ms resulted in an amplitude of only 1.83±3.42 mV (n=34, N=25) (*Figure 5F, Figure 5—figure supplement 1G and H*). Thus, when providing a CN-MDJ stimulus interval that could maximize (~+150 ms) or minimize (~+50 ms) the STO amplitude, the changes in these amplitudes

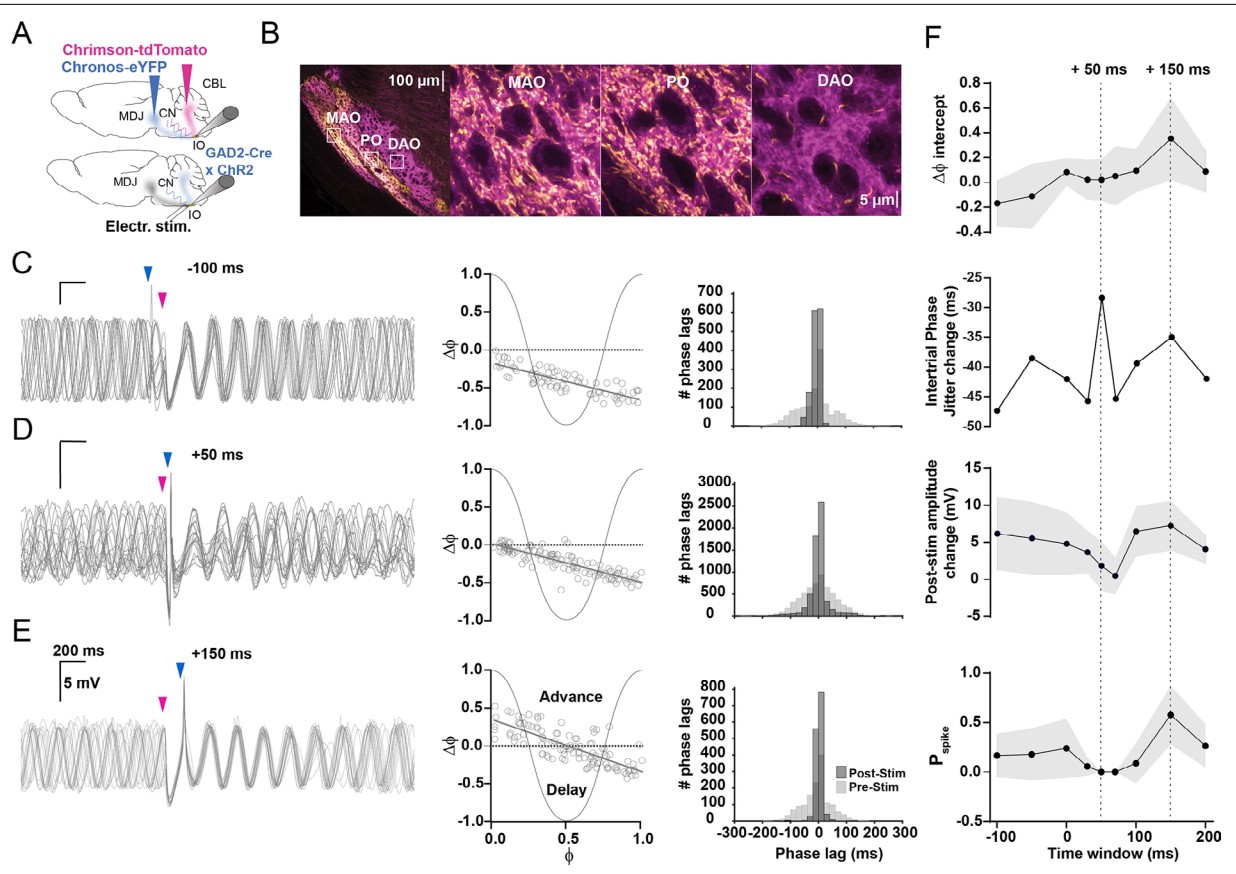

**Figure 5.** Timing of input from mesodiencephalic junction (MDJ) with respect to that from the cerebellar nuclei (CN) is critical for olivary output. (**A**) To stimulate the excitatory input from the MDJ to neurons in the principal olive (PO) and medial accessory olive (MAO), we used either optogenetic stimulation using Chronos-eYFP (top) or direct electrical stimulation (bottom panel), while we used ChrimsonR-tdTomato in wild type mice (top) and ChR2-H134R-EYFP (bottom) in Gad2Cre transgenic mice to stimulate their inhibitory input from the CN. (**B**) Confocal images of a sagittal section of the brainstem comprising the different olivary subnuclei. Note that the PO and MAO, but not the dorsal accessory olive (DAO), receive a prominent input from both the MDJ and CN. Insets on the left correspond to higher magnifications on the right. (**C**) Whole-cell patch-clamp recordings of an olivary neuron that shows subthreshold oscillations, the phase of which is influenced by the timing of the MDJ input (blue arrow-heads) with respect to that of the input from the CN (pink arrow-heads). Note that the amplitude of the oscillations is minimal and maximal when the MDJ stimulation is preceded by CN stimulation at 50 ms (middle panel) and 150 ms (top panel), respectively. (**D**) Phase plot curves of olivary activity following MDJ and CN stimulation at the three different time intervals of –100 ms (top), +50 ms (middle), and +150 ms (bottom). (**E**) Histograms showing the distribution of phase lags calculated across trials for three cycles before (pre-stimulus; light gray) and three cycles after (post-stimulus; dark gray) the sequence of MDJ and CN stimulations that were separated by the same three intervals (–100 ms in top panel, +50 ms in middle panel, and +150 ms at bottom panel). (**F**) From top to bottom, ΔΦ-intercept of phase response curves, intertrial phase jitter change (SDpost-stim – SDpre-stim), post-stimulus amplitude change of the first cycle following the last stimulation, and spike probability, all measured at time intervals of –100 ms, –50 ms, 0 ms, +30 ms, +50 ms, +70 ms, +100 ms, +150 ms, and +200 ms. Dashed lines highlight +50 ms and +150 ms time intervals. Note that the jitter is highest when the inhibition is greatest, that is, at the MDJ-CN interval of +50 ms, and that the spike probability is highest when the ΔΦ-intercept and post-stimulus amplitude change are greatest, that is, at the MDJ-CN interval of +150 ms. Shaded areas represent SD.

The online version of this article includes the following figure supplement(s) for figure 5:

**Figure supplement 1.** Phase response curve (PRC) calculation.

were larger than when presenting CN inhibition or MDJ excitation in isolation (p<0.001 for both comparisons, Kruskal-Wallis test) (*Figure 5F*, *Figure 5—figure supplement 1G and H*).

Taken together, our data demonstrate that specific temporal interactions of inhibitory and excitatory postsynaptic responses in IO neurons dictate the phase of their STOs, ranging from inconsistent variable resets to reproducible phase entrainments, including delaying as well as advancing effects. The possibility to vary the intervals between CN inhibition and MDJ excitation enriches the input-output repertoire of the olivary system in that the maximum potential effects, be it on the phase value or the amplitude, are always greater than when giving CN inhibition or MDJ excitation in isolation.

## CN input and IO spiking

In awake animals the spontaneous firing rate of complex spikes is approximately 1 Hz (*Ju et al., 2019*). Yet, in our slice preparation of the IO 57% of the cells analyzed (32 out of 56 neurons, N=36) showed spontaneous spikes at a significantly lower frequency of 0.1±0.1 Hz (ranging from 0.0017 to 0.34 Hz, membrane voltage range: from –56 to –41 mV). The spontaneous firing frequency of olivary cells in vitro depended on the membrane potential in that firing frequency tended to increase as membrane voltage was more depolarized (firing frequency from –56 to –51 mV: 0.057±0.066 Hz, n=9; N=9; from –51 to –46 mV: 0.083±0.086 Hz, n=18, N=15; from –46 to –41 mV: 0.20±0.11 Hz, n=5, N=5).

In general, CN stimulation exerts a strong hyperpolarizing effect in IO cells (*Figure 2*), preventing spike generation. Indeed, in most cells (6 out of 7) an additional depolarizing current step (580.1±293.9 pA) was required to reduce the membrane potential to an average of –40.24±9.52 mV

**Figure 6.** Impact of cerebellar nuclei (CN), mesodiencephalic junction (MDJ), and combined CN-MDJ stimulation on suprathreshold responses of inferior olive (IO) oscillating neurons. (**A**) Suprathreshold responses in IO neurons evoked by CN stimulation. In the left panel CN stimulus onset is indicated by a pink arrow-head, while action potentials are clipped in order to visualize the induced phase resetting. Bottom part of left panel indicates rising (orange) and decaying (purple) phase of subthreshold oscillations (STOs) to visualize the consistency across trials. Phase response curves (PRCs) of suprathreshold responses of individual IO cells (open circles) to CN stimulation are shown in the second column on the left. Histogram in the middle panel shows the distribution of phase lags of the STOs after the spikes calculated across trials for three cycles before (pre-stimulus) and after (post-stimulus) CN stimulation. Fourth panel shows the changes in STO amplitude following CN stimulation (for each cycle the average of the pre-stimulus cycles was subtracted). Shaded area represents SD. Fifth panel, that is, panel on the right, shows the probability that an olivary spike occurs in the first cycle directly after a preceding spike. (**B**) As in A, but now for suprathreshold responses in IO neurons evoked by MDJ stimulation. MDJ stimulus onset in the left panel is indicated by a blue arrow-head. (**C**) As in A and B, but now for suprathreshold responses in IO neurons evoked by sequential CN-MDJ stimulation with a 150 ms interval. CN and MDJ stimulus onsets in the left panel are indicated by pink and blue arrow-heads, respectively. Note that the change in the amplitude of the STO as well as the spike probability is higher following this 150 ms interval than when the CN and MDJ stimulation is given in isolation.

(n=6, N=6) and elicit one or more spikes. Only in one instance (1 out of 7) we could trigger spiking activity in the absence of a depolarizing current step; this event, which occurred at a membrane potential of –56.25±0.52 mV, probably resulted from a relatively strong secondary rebound component in the inhibitory synaptic response (*Figure 6A*). The spikes driven by CN afferent stimulation (most of them in conjunction with a depolarizing current step) exhibited PRCs with a slope (–0.54±0.04, $r^2$=0.78; n=8, N=7) similar to that of the subthreshold responses to CN stimulation (p=0.68, ANCOVA), but their $\Delta\Phi$-intercepts (0.15±0.02, n=8, N=7) were significantly different (p<0.0001, ANCOVA) in that the inhibition-triggered spikes could cause both phase advances and delays (compare second panel of *Figure 6A* with *Figure 2G*). The change in intertrial phase jitter of spiking activity after CN stimulation ($\Delta$SDpost-pre-CN_supra = –22.95; n=7, N=6) was significantly different (p<0.0001, F test) from that of the CN stimulation evoking subthreshold synaptic responses ($\Delta$SDpost-pre of CN subthreshold stimulation = –39.16). Inhibition-driven suprathreshold responses like their subthreshold counterparts increased the post-stimulus STO amplitude, decrementing sequentially across the consecutive cycles ($\Delta$STO amplitude first cycle: 4.04±3.19 mV, p<0.0001; second cycle: 2.89±2.79 mV, p<0.0001; third cycle: 2.72±3.04 mV, p<0.0001; fourth cycle: 2.5±2.93 mV, p<0.0001; n=7, N=6, Friedman test) (*Figure 6A*). Thus, the first cycle showed the largest change in amplitude in line with the post-stimulus rebound observed during subthreshold responses (compare *Figure 6A* with *Figure 2I*).

## MDJ input and IO spiking

Unlike subthreshold excitatory responses (*Figure 3G*), suprathreshold spikes driven by MDJ afferent stimulation caused mostly phase delays distributed along a clear PRC slope (–0.50±0.02, $r^2$=0.82; p=0.4127; n=6, N=5, ANCOVA), (*Figure 6B*). When comparing suprathreshold responses to MDJ and CN afferent stimulation, the PRC slopes were comparable (p=0.40, ANCOVA), but the $\Delta\Phi$-intercept was significantly shifted downward for MDJ stimulation (p=0.0093, ANCOVA). Thus, spikes elicited by either MDJ or CN afferent stimulation give rise to a similar relationship between the phase at which the input is triggered and the phase change that is induced, but in contrast to spikes triggered by CN inhibition, MDJ spikes cause more phase delays with post-stimulus cycle periods that are usually longer than those prior to stimulation.

As reflected by the PRC data (*Figure 6B*) and in line with spontaneously occurring spikes (*Bazzigaluppi et al., 2012a*), spikes triggered by MDJ stimulation usually occurred around the peak of the oscillation. Action potentials caused a relatively mild, yet significant, reduction in the intertrial phase jitter (SDpost-stim=36.05 ms versus SDpre-stim=59.10 ms; p<0.0001, F test, n=8, N=7) (Figure 6—supplementary file 1E). Moreover, suprathreshold MDJ afferent stimulation also significantly increased the post-stimulus oscillation amplitude at multiple cycles ($\Delta$STO amplitude first cycle = 6.75 ± 4.26, p<0.0001; second cycle = 1.76 ± 2.24, p<0.0001; third cycle = 1.15 ± 1.63 mV, p<0.0001; and fourth cycle = 0.73 ± 1.59, p<0.0001; n=10, N=9, Friedman test) (*Figure 6A and B*). The increases in post-stimulus oscillation amplitude following suprathreshold MDJ stimulation were significantly larger than those induced by subthreshold excitatory MDJ input (first cycle, p<0.0001; second cycle, p<0.0001; third cycle, p<0.0001; fourth cycle, p=0.0123; Mann-Whitney U test), CN subthreshold input (first cycle, p<0.0001; second cycle, p<0.0001; third cycle, p<0.0001; fourth cycle, p=0.03; Mann-Whitney U test), as well as CN suprathreshold input (first cycle, p<0.001; second cycle, p=0.06; third cycle, p=0.013; fourth cycle, p<0.0001; Mann-Whitney U test).

Finally, we compared the shape and properties of olivary spikes generated by MDJ afferent stimulation with those triggered by CN afferent stimulation in non-oscillating cells. We selected spikes for each condition evoked with the same current pulses (760±106.9 pA, range: 600–900 pA, for both conditions) and a comparable average membrane potential prior to stimulation (CN spikes: –36.26±3.75 mV, n=4, N=4; MDJ spikes: –36.92 mV ±4.86 mV, n=4; p=0.30, paired t-test). Spikes driven by MDJ and CN inputs exhibited a comparable half-width (MDJ spikes: 21.48±4.04 ms n=4, N=4; CN spikes: 22.02±5.88 ms n=4, N=4; p=0.70, paired t-test). MDJ-triggered spikes showed a trend to have a reduction in the number of spikelets as compared to CN-triggered spikes, but this trend did not reach significance (MDJ spike: 2.06±2.27, n=4, N=4; CN spike: 3.19±2.32, n=4, N=4; p=0.09, paired t-test). Furthermore, MDJ spikes showed a lower rheobase (MDJ spikes: 287.5±201.6 pA, n=4, N=4; CN spikes: 675±50 pA, n=4, N=4; p=0.018, paired t-test), highlighting their impact on the excitability of the cells (*Bazzigaluppi et al., 2012a*; *Mathy et al., 2014*).

## Spiking responses to closely timed CN and MDJ inputs and impact on network size

Next, we addressed how spike probability is affected by the relative timing of inhibition and excitation with respect to each other. As expected, spike probability was low when excitation followed inhibition at intervals in the range of 50–70 ms (Pspike = 0 ± 0 in both cases; for 50 ms, n=9, N=8; for 70 ms, n=6, N=6), both near the trough of the hyperpolarizing component of the inhibitory response (*Figures 5F and 6C*). Spike probability was largest at an interval of 150 ms around the peak of the rebound component of the inhibitory response (0.57±0.3; n=9, N=8) (*Figures 5F and 6C*, *Figure 5—figure supplement 1I and J*), which coincides with the major resonant frequency of IO cells (*Khosrovani et al., 2007*; *Negrello et al., 2019*). The spike probability for the 150 ms interval was significantly higher than for either single inhibitory (0.01±0.03, n=7, N=6; *Figure 6A*) or excitatory (0.22±0.28, n=7, N=6; *Figure 6B*) responses under the same Vm and stimulation strength (150 ms interval with respect to single inhibitory and single excitatory responses p<0.0001 and p=0.0018, respectively; one-way ANOVA) (*Figures 5F and 6C*, *Figure 5—figure supplement1I and J*).

The specific interval between inhibition and excitation also appeared to prominently affect the amplitude of the network oscillations. We tested this in the realistic large-scale network model of the MAO highlighted above (see also *Negrello et al., 2019*). Aiming at a realistic reconstruction of the interaction between incoming arborizations, model neurons were placed in a network according to locations derived from anatomical annotations of the MAO (*Vrieler et al., 2019*; Marylka Uusisaari, Nora Vrieler, and Nick Medvedev, unpublished data made available with the model). In the network, gap junctions between cells in the olivary network were established probabilistically as a function of proximity where cells within a radius of 120 μm were allowed to make connections. Conductances

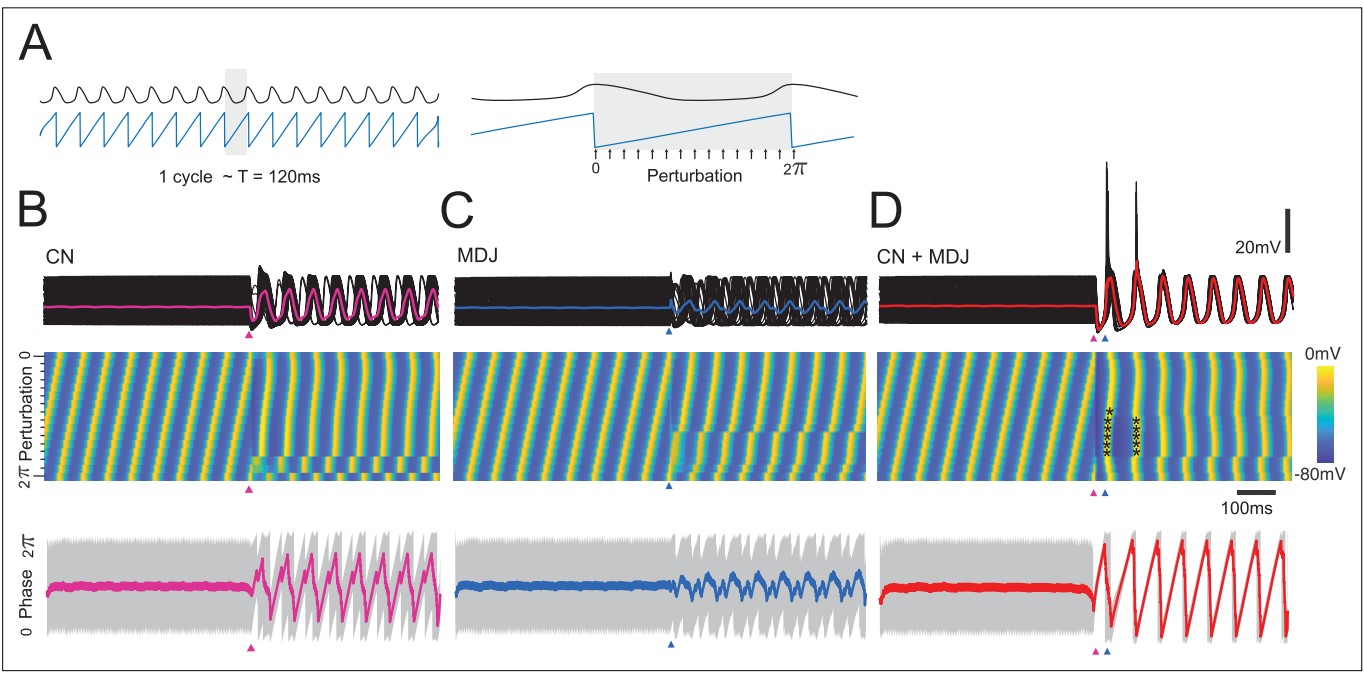

**Figure 7.** Modeling indicates that resetting of subthreshold activity and spike generation are more effective at the preferred interval. A single model example cell (with a Ca v.3.1 conductance of 1.1 mS/cm²) is stimulated at different phases of the oscillation. Stimulations were delivered at 16 different moments within one cycle. (**A**) The cell is stimulated in different sub-intervals of a single oscillation. Responses are given with respect to subthreshold inhibition (**A**), excitation (**B**), or combined stimuli (t(exc)-t(inh)=100 ms) (**C**). Traces are aligned by stimulus (and last stimulus in the case of combined stimulation). Top panel indicates membrane potential traces for all stimulation intervals; middle panel indicates membrane potential heatmap and individual traces of membrane potential sorted by increasing phase; and the bottom panel indicates the phase averages (thick line) and confidence interval (gray). The phase reset is most effective when inhibition (GABA receptor activation) and excitation (AMPA receptor activation) happen at the preferred interval of about one cycle. GABA or AMPA activation alone show a weaker resetting and produce a larger spread of phases than combined inhibition and excitation. Note that at the parameter settings used for this graph, the preferred interval between cerebellar nuclei (CN) and mesodiencephalic junction (MDJ) stimulation tends to evoke sodium spikes and rebounds (indicated by dots) for many phase values of the subthreshold oscillation (STO), whereas isolated stimulation of CN or MDJ does not.

were randomized (*Figure 8—figure supplement 1*) and the resulting network was tuned to reproduce the observed proportion of oscillators (*Figure 9—figure supplement 1*).

When we studied the amplitude of oscillations and phase differences between stimulated and non-stimulated groups, we observed the same features as observed in the experiments; indeed, the characteristic features emerged from both simulations of single cells (*Figure 7*) and of the entire murine IO (*Figures 8 and 9*). Resetting of subthreshold activity and spike generation were more prominently affected at the preferred interval where MDJ excitation occurred one STO cycle after inhibition (note that the cycle in the model network is closer to 8 Hz; the preferred interval was 100 ms), than when inhibition or excitation occurred in isolation (compare *Figure 7B and C* with *Figure 7D*). The network responses including spiking probability and phase resetting also closely followed the pattern of changes in amplitude of the STOs in that the lowest and highest levels were reached with the combinations of the inhibitory and excitatory inputs occurred at the appropriate cycle intervals (*Figures 8 and 9*). To examine the result of interactions between converging GABAergic and AMPA inputs in detail, we swept over a set of intervals between inhibition and excitation pulses and recorded amplitude of rebound and synchrony levels over time. When focal stimulation of the excitatory input to a group of cells occurred one cycle after inhibition onset (cycle T=120 ms, 8 Hz), the network produced the highest amplitude of rebound, largest spiking probability and effective phase resets (*Figure 8—figure supplement 3*). For most stimulus intervals, combined stimulation detached the oscillation of the stimulated group, which could last for over a second (*Figure 8C and D*, *Figure 8—videos 1 and 2*). This effect was also evident when comparing the evolution of phase distributions of cells in the network for the different intervals (*Figure 8—figure supplement 2*). When network cells received a modicum of stochastic input (i.e., according to an Ornstein Uhlenbeck process, with a mean of 0 pA and an SD of 0.2 pA), the resetting effects on the oscillations were less prominent (*Figure 8—figure supplement 3*, *Figure 8—video 3*), indicating that a network under normal awake conditions is unlikely to be synchronous for longer periods. In effect, stimulation with different intervals can create clusters with sub-cycle phase differences across the stimulated group of cells and the rest of the network (*Figure 8D*, *Figure 8—video 1*), highlighting the rich temporal and spatial patterns of IO activity that may arise from all its possible combinations of inhibitory and excitatory inputs.

## Discussion

The timing of the output of neurons of the IO is critical for the role of the olivocerebellar system in both short-term online control of sensorimotor integration (*Welsh et al., 1995*; *De Gruijl et al., 2014a*; *De Gruijl et al., 2014b*; *Hoogland et al., 2015*) and long-term induction of motor learning (*Ito and Kano, 1982*; *Raymond and Lisberger, 1998*; *Romano et al., 2018*; *Romano et al., 2022*; *Bina et al., 2022*; *De Zeeuw, 2021*). It has been proposed that this timing does not only depend on the intrinsic oscillatory properties of IO neurons (*Llinás and Yarom, 1981*; *Llinás and Yarom, 1986*; *Lampl and Yarom, 1993*; *Lampl and Yarom, 1997*; *Choi et al., 2010*; *Turecek et al., 2016*), but also on the activity of their inhibitory and excitatory afferents, both of which ubiquitously target dendritic shafts and spines in the olivary neuropil (*De Zeeuw et al., 1998*). Here, we present the first systematic investigation on the effects of combined inhibition and excitation on STOs and spike probability of IO neurons, elucidating how inhibitory GABAergic afferents from the CN and excitatory glutamatergic afferents from the MDJ together control phase and amplitude of these oscillations and thereby their spike timing. Our data show that stimulation of the inhibitory input generates a powerful phase resetting of the STOs of IO neurons, and that subsequent stimulation of their excitatory input one oscillation cycle later optimizes the probability to maximally depolarize the cell and generate a spike. In contrast, excitation following inhibition by only several tens of milliseconds minimizes depolarization as well as spike generation. These maximum and minimum spike probabilities following combined stimulations of the inhibitory and excitatory olivary inputs at specific intervals are relatively high and low, respectively, highlighting the sensitivity to and relevance of interval timing of both inputs in olivary gate control.

### Olivary responses to CN and MDJ stimulation

Our data elucidate how activation of afferents presented to the IO, either independently or jointly at varying intervals, affect STO amplitude, phase and spike probability. Inhibitory sub- and suprathreshold

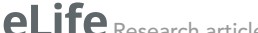

**Figure 8.** Network model of inferior olive can recapitulate timing-sensitive interaction between inhibitory and excitatory inputs. Combined stimulation at a preferred interval of about one cycle leads to highest rebound and maintenance of network synchronization. (**A**) Data from model of the olivary neuropil based on a reconstruction of the mouse's medial accessory olive (MAO). The model comprises 1105 inferior olivary model cells (gray and

*Figure 8 continued on next page*

*Figure 8 continued*

colored circles), located according to annotated cell positions of MAO cells from confocal slice data. Pink colored cells indicate the cells (~80) that are solely inhibited by the cerebellar nuclei (CN) during the period of focal stimulation of the afferents; blue colored cells indicate the cells that are only activated by a similar focal excitatory input from the mesodiencephalic junction (MDJ) during the same period; and the red colored cells indicate cells that are activated by both the inhibitory input from the CN and excitatory input from the MDJ in this period. (**B**) Snapshots of network activity. The colors of the cells highlight the membrane potentials of the cells and they correspond to the color coding presented in panel C. The network activity starts in a stage of synchrony, and this synchrony is retained upon combined stimulation in the preferred interval (see also *Figure 8—video 1*). CN and MDJ stimulations are focal. (**C**) Membrane potential for all cells in the network over time. Depolarized (yellow) or hyperpolarized (blue) membrane potentials of clustered olivary cells responding to sequential CN and MDJ stimulation (pink and blue arrow-heads, respectively). Cells receiving combined stimulation are shown at the bottom (red dashes). (**D**) Average of membrane potentials of the stimulated group and the network. (**E**) Note high amplitude of rebound in D and maintenance of network synchrony in E. (**F**) Phase of stimulated group and network. For the preferred interval phase is maintained, though the stimulation leads to an increase of amplitude in the network response. (**G**) Phase distributions of stimulated cells (red) and rest of the network (gray). Individual cell phases are displayed as small circles. Averages are displayed as vectors from the center. The phase values of the individual cells are displayed as vectors linked to the cells with corresponding colors. For phase distributions over time, see *Figure 8—video 2*.

The online version of this article includes the following video and figure supplement(s) for figure 8:

**Figure supplement 1.** Cell conductances parameter spaces for coupled and uncoupled networks show effect of gap junction coupling in murine scale model network.

**Figure supplement 2.** Stimulation at non-preferred intervals leads to enduring phase differences.

**Figure supplement 3.** Responses of network to combined stimulation under noise.

**Figure 8—video 1.** Combined stimulation can lead to standing wave propagations.

https://elifesciences.org/articles/83239/figures#fig8video1

**Figure 8—video 2.** Different stimulation intervals lead to systematic phase differences.

https://elifesciences.org/articles/83239/figures#fig8video2

**Figure 8—video 3.** Oscillator phase in the reconstructed olive during the presence of Ornstein-Ulhenbeck noise.

https://elifesciences.org/articles/83239/figures#fig8video3

synaptic responses triggered by CN afferents and suprathreshold excitatory activity triggered by MDJ afferents affect STO amplitude and reset their phase, albeit with different effects with regard to their strength and timing; CN inhibition prominently resets to trough and suprathreshold MDJ excitation moderately resets to peak. By contrast, subthreshold excitatory synaptic responses following MDJ stimulation evoke smaller amplitude changes, weaker phase resetting, and a higher intertrial phase jitter of the STOs. In line with previous studies (*Leznik et al., 2002*; *Best and Regehr, 2009*; *Bazzigaluppi et al., 2012a*; *Bazzigaluppi et al., 2012b*; *Lefler et al., 2014*; *Turecek et al., 2014*; *Turecek et al., 2016*; *Garden et al., 2017*; *Garden et al., 2018*), we show that stimulation of either CN or MDJ afferents in vitro elicits transient STOs in a subset of non-oscillating neurons with the hyperpolarizing component of the synaptic response directly affecting the phase and amplitude of such induced oscillations. These findings may well also hold in intact animals, since cell-attached and whole-cell recordings of inferior olivary neurons in vivo (*Chorev et al., 2007*; *Khosrovani et al., 2007*; *Van Der Giessen et al., 2008*) as well as multiple single unit recordings of complex spike activity of ensembles of Purkinje cells in awake behaving animals (*Negrello et al., 2019*) indicate that a significant fraction of 'silent' olivary neurons can be 'awakened' and start to oscillate, depending on the status of the peripheral input.

When the CN and MDJ afferents are stimulated jointly, the effects are contingent on interstimulus interval, with differential effects on oscillation amplitude, resetting of phase, direction of phase change (either advanced or delayed), post-stimulus intertrial phase jitter, as well as spike probability. We tested a range of stimulus intervals and found that when afferent stimulation of excitatory MDJ input occurs around 150 ms after inhibitory CN afferent stimulation (i.e., ~1 phase cycle), the oscillation phase is strongly reset and the probability to generate an olivary complex spike (*Crill, 1970*) is significantly increased. By contrast, when MDJ afferents are stimulated 50 ms after inhibition, STOs show poor resetting with a high level of jitter and complex spike generation is suppressed. These data

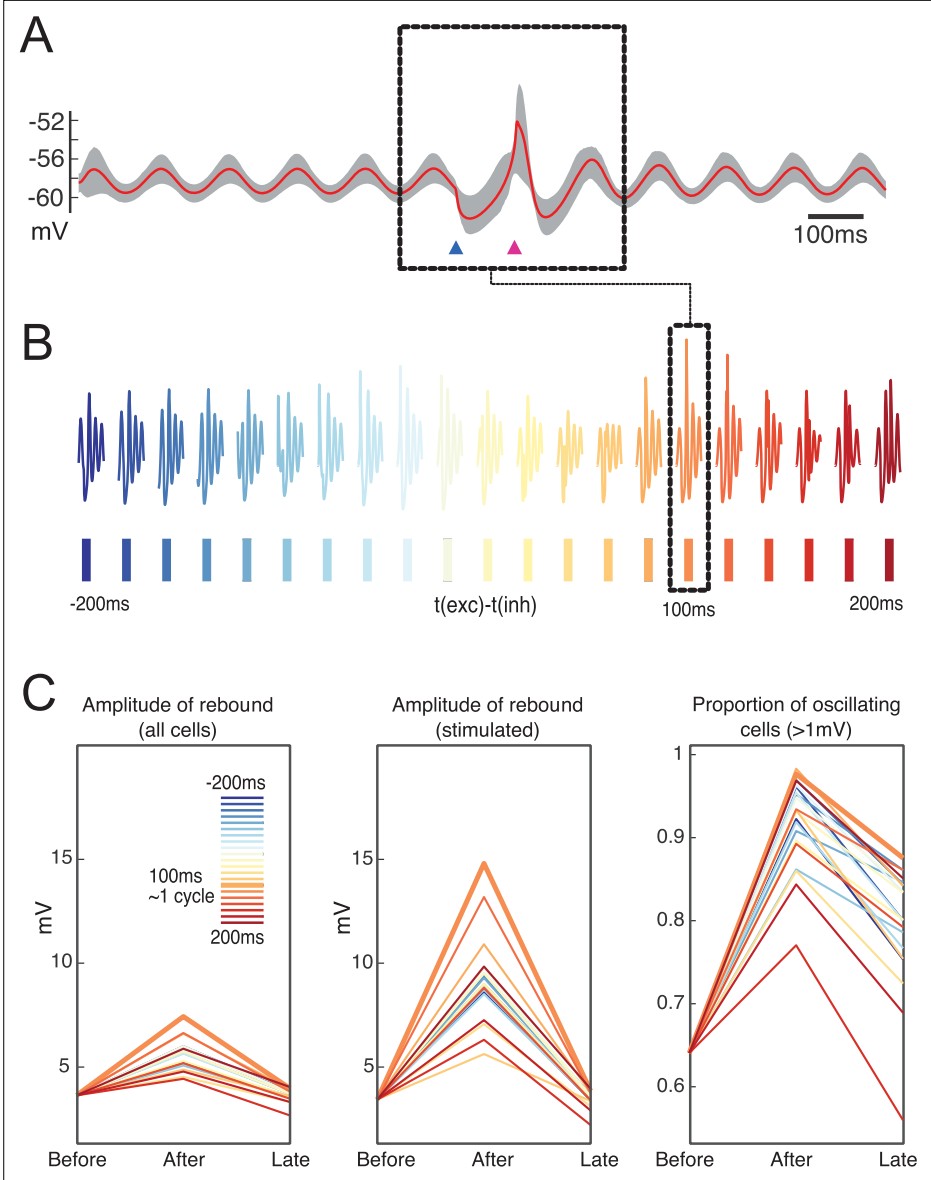

**Figure 9.** Network responses to combined stimulation for different intervals show an increase in rebound amplitude as well as an increase in the proportion of oscillating cells following stimulation in the preferred interval. (**A**) Average of membrane potential for entire network for the preferred network stimulus (100 ms). (**B**) Collected response snippets for a variety of stimulation intervals. Dashed inset indicates the interval with the largest amplitude of oscillatory response. (**C**) Comparison across network amplitude responses and proportion of oscillating cells following stimulations at different intervals (t-200 and t-100 vs. t and t+100 as well as t+400, t+500). Preferred interval is marked by thicker orange line, same as in the dashed box.

The online version of this article includes the following figure supplement(s) for figure 9:

**Figure supplement 1.** Network tuning for calcium conductance to capture proportion of oscillating cells.

suggest that the IO in general sets permissive time windows during which its afferents can suppress or facilitate complex spike generation, and that the pre-existing beat of these intrinsic windows can be overruled in a most dominant fashion when MDJ excitation follows CN inhibition by approximately 1 phase cycle of the post-CN stimulus STO.

## Prediction based on cyto-architecture of olivary neuropil

The olivary neuropil is characterized by the so-called glomeruli, which comprise a core of dendritic spines that are coupled by dendrodendritic gap junctions and surrounded by axon terminals (*Němecek*

*and Wolff, 1969*; *Bowman and King, 1973*; *Sotelo et al., 1974*). Ultrastructural 3D analysis of olivary neurons in the PO and rostral MAO following the use of a combination of intracellular labeling with double or triple labeling of afferents indicates that each and every spine of the entire dendritic arbor is innervated by both an inhibitory and excitatory input (*de Zeeuw et al., 1989*; *De Zeeuw et al., 1990*; *De Zeeuw, 1990*). Moreover, the vast majority of both the inhibitory and excitatory afferents terminates onto these spines within the glomeruli. Given that there are only very few local interneurons in the IO (*De Zeeuw, 1990*; *Fredette et al., 1992*), this dominant configuration implies that the interaction between the inhibitory and excitatory inputs onto the electronically coupled spines over time is likely to play an important role in IO function. In line with theoretical models of excitable dendritic spines that receive both an inhibitory and an excitatory input (*Segev and Parnas, 1983*; *Segev and Rall, 1988*), it has been advocated that the inputs from the CN and the MDJ onto the dendritic spines of IO neurons control the occurrence of their spikes in a highly timing-sensitive manner (*De Zeeuw, 1990*; *De Zeeuw et al., 1998*; *De Gruijl et al., 2014b*). Our cell physiological data shown here indicate that the cyto-architectonic organization of the olivary neuropil and its inputs indeed appears properly designed to generate well-timed STOs and action potentials in IO neurons, and that the phase and timing of these events strongly depend on the moments when the inhibitory and excitatory inputs are activated with respect to each other. Via their interactions over time, phase and spike timing of IO neurons can be both delayed and advanced, the reliability of the periodicity of their occurrences can be controlled, and the size of the cluster of olivary neurons in which these processes take place can be spatially regulated.

## How inhibitory or excitatory input in isolation affect spiking probability and STOs

The prominent impact of combined activation of the afferents from the CN and MDJ at the one cycle interval (i.e., CN inhibition followed by MDJ excitation one STO cycle later) in oscillating IO cells appears dominant in that the spike generation occurs, in the vast majority of trials, largely independent from the phase of the pre-existing STO at the moment of the initial inhibition provided by the GABAergic input from the CN. In fact, the combined activation at such an interval usually leads to a higher generation of spikes even when there is no pre-existing STO at all. However, the relative impact of the pre-existing phase and amplitude of the STO on spiking is greater when the MDJ afferents are stimulated in isolation. Indeed, under this condition activation of MDJ afferents virtually only leads to spiking when stimulation is presented near the peak of an oscillation (see also *Leznik et al., 2002*; *Khosrovani et al., 2007*; *Van Der Giessen et al., 2008*; *Garden et al., 2017*). Likewise, the impact of the phase of a pre-existing STO is significant when the CN afferents are stimulated in isolation in that this will determine to what extent the inhibition will delay the subsequent phase (*Figure 2G*).

The impact of CN input on STOs differs substantially from that of MDJ input. STOs induced by CN input have on average a smaller Vhyp, possibly due to the flooring effect of HCN channels, while STOs induced by MDJ input show a more depolarized resting membrane potential. Despite these differences, the STOs induced by CN input slightly, but significantly, outlast those induced by MDJ input. This might be due to the prominent impact of CN afferents on phase resetting as well as to their asynchronous release of GABA (*Best and Regehr, 2009*; *Turecek and Regehr, 2019*), which together promote a more effective resetting and thereby also persistence of oscillations. These effects are indeed also observed in the model (*Figure 7*). The relatively mild phase resetting and amplitude boost of the MDJ excitation-driven synaptic responses may be related to the conductances that are recruited during their hyperpolarizing phase. Whereas the hyperpolarizing component triggered by CN input is mediated by GABAAR-activated currents (*Devor et al., 2001*; *Lefler et al., 2014*), that triggered by MDJ input is mediated by SK and HCN channels (*Bal and McCormick, 1997*; *Marshall and Lang, 2004*; *Garden et al., 2017*; *Garden et al., 2018*). These channels might be partially inactivated in oscillating neurons during their relatively large membrane depolarizations (*Milligan et al., 2006*), reducing the impact of the excitation-mediated Vhyp.

Interestingly, the release of both GABA and glutamate may also affect the STOs indirectly by their local impact on the level of electrotonic coupling in the olivary glomeruli (*Lefler et al., 2014*; *Mathy et al., 2014*; *Turecek et al., 2014*). For example, repetitive release of the inhibitory neurotransmitter GABA from the cerebellar terminals at the olivary spines can reduce the level of coupling (*Lang et al., 1996*), possibly by shunting the spines with their relatively long necks from the dendrites (*Llinas*

*et al., 1974*; *De Zeeuw et al., 1990*; *De Zeeuw, 1990*), and thereby transiently reduce the strength of the STOs both in vitro (*Lefler et al., 2014*) and in vivo (*Lang, 2002*; *Bazzigaluppi et al., 2012a*; *Bazzigaluppi et al., 2012b*). Moreover, given the prominent glial sheaths surrounding the core of innervated and coupled dendrites inside the glomeruli (*De Zeeuw et al., 1990*; *De Zeeuw, 1990*) and given the observed long-lasting GABAergic effects on the hyperpolarization of IO neurons in both the experimental and modeling data, enduring entrapment of GABA in the glomeruli for about 60 ms may also contribute. Thus, even though a single stimulation of the CN pathway can induce or reset STOs robustly (*Figure 2*), repetitive stimulation may scramble the activity for a relatively long period, rendering a quasiperiodic oscillator network (*Negrello et al., 2019*). Likewise, depending on the duration and frequency of stimulation, the local impact of the excitatory glutamatergic afferents on the level of dendrodendritic coupling in the glomeruli varies (*Mathy et al., 2014*; *Turecek et al., 2014*), which in turn will affect the presence and strength of the STOs (*De Gruijl et al., 2014b*; *Negrello et al., 2019*). The potential impact of coupling on STOs following MDJ stimulation is supported by the finding that application of gap junction antagonists reduces excitation-mediated hyperpolarization (*Garden et al., 2018*). Thus, the impact of either CN or MDJ input on olivary oscillations is rich and dynamic, even when they do not occur conjunctively in a particular pattern.

## Cause and consequence of minimal and maximal IO firing

We show that when excitation follows inhibition by 50 ms and hits the peak of the hyperpolarizing component of the CN synaptic response, the STO amplitude boost is small, likely as a result of the depolarization-driven deactivation of HCN channels (*Bal and McCormick, 1997*; *Milligan et al., 2006*). The HCN channel conductance is activated at the trough of an oscillation and facilitates the initiation of a new STO cycle (*Garden et al., 2018*). For this interval, the weaker impact on the post-stimulus phase could derive from the weak amplitude boost of the first cycle of the oscillation. The most obvious explanation for the reduced spike probability at this interval is that excitatory synaptic responses are time-wise furthest away from spike threshold given that excitation arrives at the trough of the hyperpolarizing component of the inhibitory response. Importantly, this sequence can bring the firing frequency below that of spontaneous activity (*Figure 5F*, bottom panel).

Conversely, excitation that follows inhibition by 150 ms is most conducive to spike generation, as excitation is presented around the peak of the rebound component of the inhibitory response, which further brings the cell to spike threshold. Irrespective of whether a spike is elicited or not during an interval where excitation follows inhibition by about one cycle, larger phase shifts occur for any given phase at which the stimuli arrive, as reflected in the steeper slope of the PRC. Furthermore, in contrast to the other intervals for dual inputs or for that matter single input stimulation, which all have $\Delta\Phi$-intercept values that are either close to zero or negative and thus incur phase delays, the $\Delta\Phi$-intercept for presentation of inhibition and subsequent excitation 150 ms later shows a pronounced positive $\Delta\Phi$-intercept. Thus, following combined stimulation the phase can be either delayed or advanced depending on when nucleo-olivary and mesodiencephalic inputs hit the phase of the ongoing oscillation, thereby endowing the IO with a temporal interval that tunes phase shifts in both directions. For the same proxy one-cycle interval we find that if spikes are elicited the intertrial phase jitter is strongly reduced, whereas this phase jitter is increased in the absence of spikes. The effect of spiking on reducing phase variability might be explained by synergistic mechanisms in which decoupling caused by long-lasting inhibition is potentiated by spikes driven by MDJ inputs, which if triggered close to threshold at low firing frequency may in turn also lead to a decreased coupling strength (*Mathy et al., 2014*).

## Functional implications of timing sensitivity between inhibitory and excitatory inputs

Our findings on the effects of combined CN inhibition and MDJ excitation on IO STOs and spike probability support the idea that the close proximity of excitatory and inhibitory inputs on electrically coupled dendritic spines endow olivary neurons with a gating mechanism that allows for increases or decreases of their firing based on specific combinations of their inhibitory and excitatory inputs over time (*Segev and Parnas, 1983*; *Segev and Rall, 1988*; *De Zeeuw et al., 1998*; *De Zeeuw, 2021*). We show that inhibitory and excitatory synaptic responses have both hyperpolarizing and depolarizing components, which make them suitable to modify subthreshold and suprathreshold activity when

triggered at specific time intervals. Given that these features were also replicated in our network model and that the size of olivary clusters with sub-cycle phase differences could also be controlled by the inhibitory and excitatory inputs in this model, it is attractive to hypothesize that the IO circuitry may allow for phase-based computing that is under control of the synaptic inputs from the CN and MDJ. Such analog computing system can facilitate the generation of a complex array of properly sequenced actions (*Holden et al., 2017*), in which the sequence of events can be maintained at a variety of speeds or frequencies (*Jacobson et al., 2009*). Indeed, in line with the findings from our olivary network model, systems recordings of complex spike activity in the cerebellar cortex using chronically implanted multi-electrode arrays in freely moving rats indicate that the different IO clusters may be able to generate olivary oscillations with subtle phase differences, supporting intervals shorter than the cycle period, and to maintain these oscillations at a constant phase difference across large global changes in frequency (*Jacobson et al., 2009*). Accordingly, the patterns of complex spike activity can be related to ongoing reflex-like behavior and associative sensorimotor behavior (*Welsh et al., 1995*; *Van Der Giessen et al., 2008*; *De Gruijl et al., 2014a*; *Hoogland et al., 2015*), and disturbing these patterns at the sub-cycle range can affect the latencies of these motor behaviors (*Kistler et al., 2002*; *Van Der Giessen et al., 2008*). The IO may thus underlie not only rhythmic activity and synchronization, but also temporal patterning that requires CN-MDJ intervals at various cycle durations and that may enable the olivo-cerebellar system to gate temporal patterns at different rates without distortion, facilitating execution of the same behavioral task at different speeds (*Welsh et al., 1995*; *Ozden et al., 2012*; *De Gruijl et al., 2014a*; *Hoogland et al., 2015*; *Lawrenson et al., 2016*).

In addition to online control of temporal patterns of complex spike activity that may facilitate ongoing coordination of sensorimotor behavior and/or cognitive functions (*Welsh et al., 1995*; *Hoogland et al., 2015*; *Romano et al., 2018*; *Romano et al., 2022*), increases and decreases of complex spike activity are also critical for cerebellar motor learning in that they facilitate induction of long-term depression and potentiation at the parallel fiber to Purkinje cell synapses, respectively (*Ito and Kano, 1982*; *Schonewille et al., 2010*; *Gutierrez-Castellanos et al., 2017*) as well as opposite plasticity at the synapses of the molecular layer interneurons (*Gao et al., 2012*). Long-term depression and long-term potentiation at the Purkinje cell synapses have been proposed to be the dominant forms of plasticity in the so-called downbound and upbound microzones modules (*De Zeeuw, 2021*; *Voerman et al., 2022*), which underlie different forms of motor learning dependent on whether they require decreases and increases in simple spike activity, respectively (*Van Der Giessen et al., 2008*; *ten Brinke et al., 2015*; *Bina et al., 2022*; *Yang and Lisberger, 2014*; *Herzfeld et al., 2018*). Thus, the current finding that the precise interval between inhibition and excitation of IO neurons determines whether and when the complex spike firing frequency will go up or down is probably critical not only for online motor performance, but also for more long-term processes such as motor learning.

## Methods

**Key resources table**

| Reagent type (species) or resource | Designation | Source or reference | Identifiers | Additional information |
|---|---|---|---|---|
| Strain, strain background (*Mus musculus*) | C57BL/6J mice | Charles Rivers | IMSR_JAX:000664 | |
| Strain, strain background (*Mus musculus*) | Gad2-IRES-Cre × Ai32 RCL-ChR2(H134R)/EYFP | https://doi.org/10.1073/pnas.1017210108; https://doi.org/10.1038/nature13276 | | |
| Transfected construct (AAV1-Synapsin-Chronos-GFP) | Chronos | https://doi.org/10.1038/nmeth.2836 | | |
| Transfected construct (AAV1-Synapsin-ChrimsonR-tdTomato) | Chrimson | https://doi.org/10.1038/nmeth.2836 | | |

*Continued on next page*

*Continued*

| Reagent type (species) or resource | Designation | Source or reference | Identifiers | Additional information |
|---|---|---|---|---|
| Chemical compound, drug | Alexa Fluor 594- Conjugated Streptavidin | Life Technologies | | Stock concentration 2 mg/ml (1:200 dilution) |
| Chemical compound, drug | Alexa Fluor 633- Conjugated Streptavidin | Life Technologies | | Stock concentration 2 mg/ml (1:200 dilution) |
| Chemical compound, drug | APV | Hello Bio | #HB0225-50mM | A final concentration of 50 µM was used |
| Chemical compound, drug | CNQX | Hello Bio | #HB0205-50mg | A final concentration of 20 µM was used |
| Chemical compound, drug | Picrotoxin (PTX) | Sigma-Aldrich | #P1675-5G | A final concentration of 100 µM was used |
| Chemical compound, drug | Biocytin | Sigma-Aldrich | #B4261-50MG | A final concentration of 0.5% (wt/vol) was used |
| Software, algorithm | Clampfit 11.0.3 | Molecular Devices | | |
| Software, algorithm | Prism 9 | GraphPad | | |
| Software, algorithm | MATLAB | MathWorks | | |

## Animals

All performed experiments were licensed by the Dutch Competent Authority and approved by the local Animal Welfare Body, following the European guidelines for the care and use of laboratory animals Directive 2010/63/EU.

## GAD2-Cre × Ai32 transgenic mice

For a subset of experiments GAD2-Cre animals were crossed with the Ai32 line (#012569, Jackson Laboratory, ME, USA) expressing the ChR2(H134R)-EYFP fusion protein to allow selective light activation of inhibitory nucleo-olivary afferents.

## Viral transduction

Animals received bilateral 50–100 nl injections of AAV1-Synapsin-Chronos-GFP in the mesodiencephealic junction (bregma: –2.46 to –2.54, lateral: 0.6–0.7, depth: 3.8–4 mm) and AAV1-Synapsin-ChrimsonR-tdTomato in the CN (bregma: –6.1, lateral: 2, depth: 3.3 mm). Virus was injected using a Nanoject II (Drummond Scientific) and glass capillaries (3-000-203-G/X, Drummond Scientific) backfilled with mineral oil. The virus was injected at 24.7 nl/s four times spaced at intervals spaced 1 min apart. The pipette was slowly retracted over the course of 5 min after the last of the four injections. Injected animals were not used in experiments until at least 4 weeks after injection.

## In vitro slice electrophysiology

Brainstem sagittal sections (300 µm) were cut in ice-cold cutting solution (in mM: 252 sucrose, 5 KCl, 1.25 $NaH_2PO_4$, 26 $NaHCO_3$, 10 glucose, 0.5 $CaCl_2$, 3.5 $MgSO_4$, pH = 7.4) of mice (1–3 months) using a vibratome (Microm HM 650V, Thermo Scientific). Slices were then transferred to a slice chamber with ACSF (in mM: 126 NaCl, 3 KCl, 1.2 $KH_2PO_4$, 26 $NaHCO_3$, 10 glucose, 2.4 $CaCl_2$, 1.3 $MgSO_4$, pH = 7.4) in a water bath and kept first for 30 min at 34°C and then 30 min at room temperature prior to use. Experiments were performed at 32°C. Fire-polished patch pipettes (4–5 MΩ) were backfilled with internal solution (in mM: 140 K-gluconate, 4 NaCl, 0.5 $CaCl_2$, 5 EGTA, 4 Mg-ATP, 10 HEPES, pH = 7.2, osmolarity: 290 mOsm). Data were acquired using a multiclamp 700B patch-clamp amplifier and digitized at 50 kHz using a 1440A digidata digitizer. Giga-seals were obtained in voltage-clamp configuration. After break-in current clamp was used to monitor membrane voltage fluctuations.

Our criteria to reject cells were based on access resistance value and stability. Cells typically showed access resistance values of less than 20 MΩ (on average 14.09±8.74 MΩ (n=93) with 81% of cells ≤20 MΩ; 17%>20 MΩ and <40 MΩ; 2%>40 MΩ and <60 MΩ). If the recordings showed an access

resistance of >40 MΩ and/or the baseline that was unstable for >5 min the recording was rejected. Membrane potentials were not corrected for the electrode junction potential, the value of which was about –7 mV.

### Pharmacology
Stocks of CNQX (20 mM, Hello Bio #HB0205-50mg) and D-APV (50 mM, Hello Bio #HB0225-50mM) were prepared in water, whereas picrotoxin (100 mM, Sigma #P1675-5G) was prepared in DMSO in order to be dissolved in 100 ml of ACSF to reach concentrations of 20 µM, 50 µM, and 100 µM, respectively.

### Intracellular labeling
Neurons were labeled by adding 0.5% (wt/vol) biocytin (Sigma) into the internal solution. Once the experiment was completed, patch pipette was withdrawn carefully in order to reseal the neuronal membrane. Brainstem slices were then fixed overnight in 0.1 M PBS containing 4% PFA. After the fixing period, they were washed three times in 0.1 M PB (10 min per wash at 4°C) and then incubated with Alexa Fluor 594- or 633-conjugated streptavidin (Life Technologies, 2 mg/ml) in 0.1 M PB containing 0.6% Triton X-100. The choice of the fluorophore depended on whether the experiments were carried out in either transgenic or viral transduced animals, respectively. Finally, slices were washed three times in 0.1 M PB (10 min per wash at 4°C), mounted with Dako glycergel fluorescence mounting medium (RI 1.47–1.50; Dako) and coverslipped. Images of labeled neurons and olivary afferents were obtained using confocal microscopy (Leica SP8). High magnification image stacks were obtained with 40 or 63× plan-Apochromat objectives at high resolution (ranged from 0.11 to 0.38 µm/pixel in the XY plane and 0.1–0.5 µm steps in the Z plane).

### Electrical stimulation
Bipolar stimulation electrodes were placed at the rostral border of the olivary subnucleus in order to stimulate the descending pathways originating from the MDJ (medial and central tegmental tracts). Brief 100 µs current pulses of 0.3–1.2 mA were applied using an ISO-flex stimulus isolator.

### In vitro optogenetics
We used a three-wavelength microscopy LED system in combination with a control box (Mightex Systems, Toronto) exciting Chronos at 470 nm and ChrimsonR at 625 nm using TTL pulse triggering. Slices were illuminated through a 40× objective (NA=0.5, Zeiss).

### Data analysis
Data were analyzed using Clampfit 11.0.3 and custom scripts written in MATLAB. All data and charts are expressed as mean ± standard deviation. Membrane voltage, Vm, was determined as the average of the voltage baselines preceding the current steps used for the I-V curve (*Figure 1D*). When measuring the spontaneous spiking as a function of Vm, membrane voltage was taken as the average of a spontaneous activity trace (231±153 s, ranging from 22 to 599 s). Rn was measured as the slope of the linear part of the I-V curve. For the I-V curve, current steps were typically provided from –0.4 nA to 1 nA. The linear part of the curve was usually between –0.4 nA and 0.1 nA. Phase values are predominantly presented as measurements of the PRC (for details, see *Figure 5—figure supplement 1*) and as their lag distribution. Phases of STOs were extracted using DAMOCO MATLAB toolbox, from which phase lags were calculated based on the coupled oscillator approach (*Kralemann et al., 2008*) across trials for three cycles before and after stimulation. All details about calculations of the initial phase estimates (proto-phases) and then its conversion to phases as well as phase dynamics can be consulted in the link, http://www.stat.physik.uni-potsdam.de/~mros/damoco2.html.

In order to determine the impact of stimulation on intertrial phase jitter the standard deviation of the phase lags pre- and post-stimulus (SDpre-stim and SDpost-stim, respectively) was calculated. To compare the impact of different types of stimulation on the intertrial phase jitter, the intertrial phase change was calculated as the value of SDpost-stim minus SDpre-stim (together also referred to as ΔSDpost-pre) for each experimental condition.

## Statistics

All statistical analyses were performed in GraphPad Prism 8.3.0. In order to test whether the data was normally distributed, D'Agostino-Pearson omnibus normality test was performed. For comparisons of two groups showing normally distributed data, unpaired or paired t-test was used, otherwise, Mann-Whitney U test or Wilcoxon matched-paired rank test was performed for unpaired or paired data, respectively. When multiple normally distributed groups were compared, a one-way ANOVA test is followed by a post hoc uncorrected Fisher's LSD multiple comparison test. Otherwise, for unpaired data, Kruskal-Wallis test followed by uncorrected post hoc Dunn's test was used, or in case of paired data a Friedman test followed by post hoc uncorrected Dunn's test. To determine whether the slope and intercept of two linear regressions were significantly different, we used the ANCOVA test, while for multiple comparisons one-way ANOVA test followed by post hoc uncorrected Fisher's LSD multiple comparison test was used.

To determine the impact of stimulation on intertrial phase jitter, F test was used to compare the standard deviation pre- and post-stimulus (SDpre-stim and SDpost-stim, respectively). Moreover, to assess and compare the impact of different types of stimulation on intertrial phase jitter, F test was used to compare the values resulted from the subtraction of SDpost-stim and SDpre-stim (or ΔSDpost-pre), across different experimental conditions. For all the aforementioned tests, p-values below 0.05 were considered statistically significant.

## Computational model

The model is an extension of the network model published in *Negrello et al., 2019*. Instructions to run the scripts and produce figures can be found at README.md in the github repository https://github.com/MRIO/OliveTree (copy archived at *Negrello, 2023*). In short, the model is a reconstruction of the olivary MAO network, constituted by three-compartment Hodgkin-Huxley type cell models (soma, axon, dendrite), where model cells were placed according to annotated somata positions (kindly made available by Marylka Yoe Uusisaari, Nora Vrieler, and Nicholas Medvedev). Olivary cells in the reconstructed network followed a homogeneous connectivity. This is despite evidence that these cells are clustered, but it is a parsimonious assumption compatible with the questions of this study. We applied a probability of connectivity specifying a mean degree of 10 connections per cell. Cells were allowed to connect to cells in other neighboring cells within a radius of 120 µm. The electrotonic conductance across gap junctions of connected cells in our model was randomly picked beween 0 and 0.03 mS/cm$^2$ so that maximum coupling coefficients would be around 10% (see also *De Zeeuw et al., 1996*; *De Zeeuw et al., 2003*; *Schweighofer et al., 1999*; *De Gruijl et al., 2012*; *Latorre et al., 2013*).

Conductance parameters of the olivary cells models were randomized to create observed levels of variability in STO properties (e.g., amplitude, frequency, ADHD) and a large variety of possible neighborhoods. We used uniform random distributions within a range (rather than gaussians) to create disparate parameter ranges and highly heterogeneous cell behavior. As these are high dimensional spaces from which we draw, this results in a broad class of model neurons. The network oscillation was largely governed by the interaction between calcium channels (v 1.3 and v3.1), BK and SK channel models. Conductance ranges were selected to cover the observed variety of neuronal responses (*Bazzigaluppi and de Jeu, 2016*). Networks were tuned to produce 30–70% of intrinsically oscillating cells, covering the ranges reported in the literature (see *Figure 9—figure supplement 1*). Cell parameterizations were created with the function 'CreateDefaultNeurons.m'.

For the network inputs we designed a subset of cells in the center of the network with 'GABA' and 'AMPA' currents being delivered to the dendrites of the modeled cells (*De Zeeuw et al., 1998*). The GABA receptor conductances in soma and dendrites have been fit to reproduce the time course of activation shown by *Devor and Yarom, 2002*, while the 'AMPA' input kinetics were modeled according to *O'Donnell et al., 2011*. To represent axonal arborizations, two partially overlapping masks within a radius of 100 µm were generated for each of the afferent inputs. Within each mask 85% of the cells received synaptic pulses.

## Acknowledgements

Financial support was provided by the Netherlands Organization for Scientific Research (NWO-ALW 824.02.001; CIDZ), the Dutch Organization for Medical Sciences (ZonMW 91120067; CIDZ), Medical Neuro-Delta (MD 01092019-31082023; CIDZ), INTENSE LSH-NWO (TTW/00798883; CIDZ), ERC-adv

(GA-294775 CIDZ), and ERC-POC (nrs. 737619 and 768914; CIDZ); as well as the Dutch NWO Gravitation Program (DBI2).

## Additional information

### Funding

| Funder | Grant reference number | Author |
| --- | --- | --- |
| Netherland Organization for scientific research | NWO- ALW 824.02.001 | Chris I De Zeeuw |
| Dutch Organization for Medical Sciences | ZonMW 91120067 | Chris I De Zeeuw |
| Medical Neuro-Delta | MD 01092019-31082023 | Chris I De Zeeuw |
| INTENSE LSH-NWO | TTW/00798883 | Chris I De Zeeuw |
| ERC-adv | GA-294775 | Chris I De Zeeuw |
| ERC-POC | nrs. 737619 and 768914 | Chris I De Zeeuw |
| Dutch NWO Gravitation Program | DBI2 | Chris I De Zeeuw |

The funders had no role in study design, data collection and interpretation, or the decision to submit the work for publication.

### Author contributions

Sebastián Loyola, Mario Negrello, Conceptualization, Data curation, Formal analysis, Supervision, Investigation, Visualization, Methodology, Writing – original draft, Writing – review and editing; Tycho M Hoogland, Conceptualization, Software, Formal analysis, Supervision, Investigation, Visualization, Methodology, Writing – original draft, Writing – review and editing; Hugo Hoedemaker, Investigation, Methodology; Vincenzo Romano, Conceptualization, Supervision; Chris I De Zeeuw, Conceptualization, Supervision, Funding acquisition, Methodology, Writing – original draft, Project administration, Writing – review and editing

### Author ORCIDs

Sebastián Loyola ⓘ http://orcid.org/0000-0002-6029-2274
Tycho M Hoogland ⓘ http://orcid.org/0000-0002-7444-9279
Vincenzo Romano ⓘ http://orcid.org/0000-0002-4449-6541
Mario Negrello ⓘ http://orcid.org/0000-0002-8527-4259
Chris I De Zeeuw ⓘ http://orcid.org/0000-0001-5628-8187

### Ethics

All performed experiments were licensed by the Dutch Competent Authority and approved by the local Animal Welfare Body, following the European guidelines for the care and use of laboratory animals Directive 2010/63/EU.

### Decision letter and Author response

Decision letter https://doi.org/10.7554/eLife.83239.sa1
Author response https://doi.org/10.7554/eLife.83239.sa2

## Additional files

### Supplementary files

• Supplementary file 1. Reponse properties of neurons in the inferior olive. (A) Kinetic properties of cerebellar nuclei (CN) and mesodiencephalic junction (MDJ) synaptic responses using different stimulation approaches. Latency and half-width of hyperpolarization (Vhyp) and rebound (Vreb) synaptic components of CN synaptic responses evoked in mice transduced with ChrimsonR-tdTomato in the CN and GAD2Cre/Chr2-H134R-EYFP transgenic mice line, and latency and half-

width of depolarization (Vdep), hyperpolarization (Vhyp), and rebound (Vreb) synaptic components of MDJ synaptic response evoked in mice transduced with Chronos in the MDJ and electrical stimulation. (B) Phase response curve (PRC) parameters of different stimulation paradigms. Slope, Y-intercept, and $R^2$ of PRCs generated by dual (CN and MDJ afferents stimulation) stimulation at different time intervals (all of them evoking subthreshold synaptic responses except for '+150 ms spike' which evokes suprathreshold synaptic responses), CN afferent stimulation evoking subthreshold and suprathreshold synaptic responses ('IPSP' and 'CN spike', respectively), and MDJ afferent stimulation evoking suprathreshold synaptic responses ('MDJ spike'). (C) Statistics of PRC slopes of different stimulation paradigms. Comparison of PRC slopes between +150 ms and the other time intervals (all of them evoking subthreshold synaptic responses except for '+150 ms spike' which evokes suprathreshold synaptic responses), CN afferent stimulation evoking subthreshold and suprathreshold synaptic responses ('IPSP' and 'CN spike', respectively) and MDJ afferent stimulation evoking suprathreshold synaptic responses ('MDJ spike'), using one-way ANOVA test followed by post hoc uncorrected Fisher's LSD multiple comparison test. (D) Statistics of PRC Y-intercept. Comparison of PRC Y-intercepts between +150 ms and the other time intervals (all of them evoking subthreshold synaptic responses except for '+150 ms spike' which evokes suprathreshold synaptic responses), CN afferent stimulation evoking subthreshold and suprathreshold synaptic responses ('IPSP' and 'CN spike', respectively) and MDJ afferent stimulation evoking suprathreshold synaptic responses ('MDJ spike') using one-way ANOVA test followed by post hoc uncorrected Fisher's LSD multiple comparison test. (E) Intertrial phase jitter at different stimulation paradigms. Intertrial phase jitter pre- and post-stimulus (standard deviation of phase lags pre- and post-stimulus, respectively) and intertrial phase jitter change (standard deviation of phase lags post stimulus – standard deviation of phase lags pre-stimulus) at different time intervals of stimulation (all of them evoking subthreshold synaptic responses except for '+150 ms spike' which evokes suprathreshold synaptic responses), CN afferent stimulation evoking subthreshold and suprathreshold synaptic responses ('IPSP' and 'CN spike', respectively) and MDJ afferent stimulation evoking subthreshold and suprathreshold synaptic responses ('EPSP' and 'MDJ spike', respectively). (F). Statistics of amplitude of first cycle following stimulation at different time intervals. Comparison between subthreshold oscillation (STO) amplitude change of the first cycle (rebound component) following stimulation at different time intervals (all of them evoking subthreshold synaptic responses) and STO amplitude previous to stimulation using Friedman test followed by post hoc uncorrected Dunn's multiple comparison test. (G) STO amplitude change of first cycle following stimulation at different stimulation paradigms. STO amplitude change of the first cycle (rebound component) following dual stimulation at different time intervals (all of them evoking subthreshold synaptic responses except for '+150 ms spike' which evokes suprathreshold synaptic responses), CN afferent stimulation evoking subthreshold and suprathreshold synaptic responses ('IPSP' and 'CN spike', respectively), and MDJ afferent stimulation evoking subthreshold and suprathreshold synaptic responses ('EPSP' and 'MDJ spike', respectively). (H) Statistics of the comparison between first cycle post-stimulus change evoked by +150 ms and other stimulation paradigms. Comparison of first cycle post-stimulus change between the time interval of +150 ms and the other time intervals (all of them evoking subthreshold synaptic responses except for '+150 ms spike' which evokes suprathreshold synaptic responses), CN afferent stimulation evoking subthreshold and suprathreshold synaptic responses ('IPSP' and 'CN spike', respectively), and MDJ afferent stimulation evoking subthreshold and suprathreshold synaptic responses ('EPSP' and 'MDJ spike', respectively), using Kruskal-Wallis test followed by post hoc uncorrected Dunn's multiple comparison test. (I) Spike probability at different stimulation paradigms. Spike probability (Pspike) following dual stimulation at different time intervals and isolated CN and MDJ afferents stimulation ('IPSP' and 'MDJ', respectively).

- MDAR checklist

## Data availability

All data and codes used for main and supplemental figures have been uploaded to DRYAD.

The following dataset was generated:

| Author(s) | Year | Dataset title | Dataset URL | Database and Identifier |
|---|---|---|---|---|
| Loyola S, Hoogland TM, Hoogland TM, Negrello M, Hoedemaker H, Romano V, De Zeeuw CI | 2023 | How inhibitory and excitatory inputs gate output of the inferior olive | https://dx.doi.org/10.5061/dryad.0zpc8672v | Dryad Digital Repository, 10.5061/dryad.0zpc8672v |

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
