## [Editor Report]

Inferior olivary neurons drive complex spiking activity in Purkinje neurons of the cerebellar cortex, ultimately playing critical roles in controlling motor coordination and plasticity. Using transgenic mice or optogenetic techniques to independently control a major excitatory and inhibitory pathway to the inferior olive, the authors show solid evidence that the probability and phase of olivary neuron output depend critically on the relative timing of excitation and inhibitory inputs. Network models predict that appropriately timed excitatory and inhibitory input patterns efficiently synchronize larger clusters of inferior olivary neurons, raising the possibility that input timing can gate the output of the motor commands. These valuable findings have the potential to impact the field's understanding of sensorimotor processing.

---

## [Decision Letter]

**Decision letter after peer review:**

Thank you for submitting your article "How inhibitory and excitatory inputs gate output of the inferior olive" for consideration by *eLife*. Your article has been reviewed by 3 peer reviewers, and the evaluation has been overseen by Nace Golding, a Guest Reviewing Editor, and Laura Colgin as the Senior Editor. The reviewers have opted to remain anonymous. We thank you sincerely for your patience during the unusually long review process, which was caused by illness (as we mentioned in an earlier email).

Essential revisions:

1) Some additional analyses and figures are necessary to support the major conclusions. Some missing details also need to be included:

a) Details are needed about how Vm and Rn were measured and compared among cells.

b) The rebound from hyperpolarization should be plotted as a function of the voltage of the prior hyperpolarization.

c) In figures 2C and 3C there are no scale bars for either x or y axes.

d) For Suppl Figure 1, explain whether the synaptic blockers were added sequentially or whether the responses shown are from only a single blocker.

e) A figure should be provided to show the effect of PTX on CN inputs. An explanation of why 100uM PTX didn't fully block the inhibition should be included.

f) Figures 7 and 8 are difficult to understand and need to be clarified.

g) The health criteria for cells and the quality of recordings need further explanation.

h) The authors also need to present the criteria for rejecting recordings on the basis of health or the value of series resistance compensation. The authors should explain whether there was a correlation between average membrane potential and the magnitude of intrinsic oscillations for the relevant cells.

i). Related to the previous comment, the authors should report in the Methods whether membrane potentials reported are corrected for the electrode junction potential, as well as the value of that potential.

2) A discussion of how the results compare to natural firing patterns in vivo and natural physiological conditions would likely increase the impact of the paper.

3) Other text revisions are suggested to improve the clarity of the presentation of results.

Please refer to the individual reviews below for further details about essential revisions.

*Reviewer #1 (Recommendations for the authors):*

My main suggestion is to remove figures 7 and 8, which are difficult to follow (Figure 8) and somewhat not in line with the experimental results. In figure 7C an interval of 50ms seems to be extremely efficient in activating the neuron and resetting the oscillations, which is not in line with the main experimental (Figure 5D) or theoretical (Figure 4) results. Figure 8 is difficult to follow and its main input to the study is not clear. Is it the change in the number of affected neurons as a function of the interval?

*Reviewer #2 (Recommendations for the authors):*

This study combines the use of transgenic mice, optogenetic approaches, and patch-clamp recordings in slices to examine the interplay between specific excitatory and inhibitory synaptic inputs and intrinsic oscillations in neurons of the inferior olive. Through analyses of the response phase and amplitude, the authors show convincingly that excitation has an especially powerful effect on response amplitude and firing when it occurs during post-inhibitory rebound. These findings build on many previous studies of this circuitry and are perhaps not surprising from a mechanistic point of view. However, the value of the work is derived from both the specificity achieved in selectively activating known sources of inhibition and excitation and also the thorough and systematic analyses the authors employ to examine the complex interactions between synaptic inputs and intrinsic voltage-gated conductances. Overall the study could be better supported in places, as documented below but provides a significant advance in understanding the circuitry underlying sensorimotor coordination.

1. The presentation of health criteria for cells and the quality of recordings needs improvement. Membrane potentials are reported for both oscillating and non-oscillating cells, including an average and a range. This is a straightforward measurement for non-oscillating cells but not for oscillators. Presumably, the authors are simply averaging out the oscillations over a long time frame but this should be spelled out explicitly. Resting potentials of -40 mV would seem to be quite depolarized for a healthy neuron for either category and the variation in Vm across cells (nearly 27 mV) seems enormous.

2. The authors need to present the criteria for rejecting recordings on the basis of health, or the value of series resistance compensation, both of which are unaddressed currently. It would also be important to report whether there was a correlation between average membrane potential and the magnitude of intrinsic oscillations for the relevant cells. It would be expected that the average Vm of cells would strongly affect the relative balance between excitatory and inhibitory currents as well as their interactions with voltage-gated ion channels.

3. Related to the previous comment, the authors should report in the Methods whether membrane potentials reported are corrected for the electrode junction potential, as well as the value of that potential.

4. It would be helpful in the Discussion if there was more attention paid to the natural firing patterns of inputs, and what physiological conditions might produce different time separations between excitatory and inhibitory inputs.

*Reviewer #3 (Recommendations for the authors):*

The oscillations in these cells are known to be highly voltage-dependent due to T-type Ca channels and h channels. Therefore, there is no surprise for the effects of inhibitory and excitatory inputs in altering the STOs. The main novelty of the findings in this paper involves the quantification of how the inputs affect the IO neurons. That said, there are numerous questions that the authors should address, some minor and some more significant.

Specific suggestions:

1. How were Vm and Rn measured? No details are given. Because both CaT and h channels are active at rest, how they were measured and then compared among cells is very important.

2. Although presumably CN and MDJ inputs are monosynaptic, some mention of this and whether there are any connections among IO neurons other than electrical would be helpful for readers outside of the field.

3. The rebound from a hyperpolarization is highly voltage-dependent. You can't just measure the amplitude from the baseline without knowing the voltage at the baseline and the voltage of the prior hyperpolarization. The rebound should be plotted as a function of the voltage of the prior hyperpolarization.

4. In figures 2C and 3C there are no scale bars for either x or y axes.

5. In Suppl Figure 1, were the synaptic blockers added sequentially and thus additive, or are the responses shown from only a single blocker?

6. No figure was provided for the effect of PTX on CN inputs. This should be included. Also, why didn't 100uM PTX fully block the inhibition? Some ideas for this would be useful.

[Editors' note: further revisions were suggested prior to acceptance, as described below.]

Thank you for resubmitting your work entitled "How inhibitory and excitatory inputs gate output of the inferior olive" for further consideration by *eLife*. Your revised article has been evaluated by Laura Colgin (Senior Editor).

The manuscript has been improved but there are some remaining issues that need to be addressed, as outlined below:

*Reviewer #3 (Recommendations for the authors):*

I have even less confidence in the rigor of the electrophysiology than I did in the previous version. I asked:

1. For details on Vm and Rn. The response for Vm was ok, but not Rn. Where is the linear portion of the I-V curve (actually it should be called a V-I) curve? No V-I is shown so I have no idea whether there was a linear portion and what the rest of the V-I plots look like.

2. Details about rebound being v. dependent. They now show plots, and indeed rebound varied 2-4 fold depending on Vm, but as far as I can tell, none of this was used in their analyses such as in Figure 2 and elsewhere.

3. I asked about 100uM PTX. They now show data from what appears to be one cell but claim that "reanalysis" indicated a complete block in 5 of 5. Where are the data?

---

## [Author Response]

Essential revisions:1) Some additional analyses and figures are necessary to support the major conclusions. Some missing details also need to be included:a) Details are needed about how Vm and Rn were measured and compared among cells.

These details are now provided in the Results (Figure 1D) as well as the Methods sections. In short, membrane voltage, Vm, was determined as the average of the voltage baselines preceding the current steps used for the I-V curve. When measuring spontaneous spiking as a function of Vm, membrane voltage was taken as the average of a spontaneous activity trace. Input resistance, Rn, was measured as the slope of the linear part of the I-V curve.

b) The rebound from hyperpolarization should be plotted as a function of the voltage of the prior hyperpolarization.

This has now been included in new Figure 2—figure supplement 2

c) In figures 2C and 3C there are no scale bars for either x or y axes.

The scale bars have been added to these figure panels.

d) For Suppl Figure 1, explain whether the synaptic blockers were added sequentially or whether the responses shown are from only a single blocker.

This information has now been added to the legends of Suppl. Figure 1 (now Figure 3—figure supplement 1). Drugs were added sequentially: first PTX, then CNQX, and finally, if possible, also APV. PTX and CNQX were added in all cells (8 in total), whereas APV could only be successfully added in 4 out of 8 cells, due to the relatively long duration of the required recordings (>80 mins).

e) A figure should be provided to show the effect of PTX on CN inputs. An explanation of why 100uM PTX didn't fully block the inhibition should be included.

We now added a Suppl. Figure (Figure 2—figure supplement 1) showing a full blockage of inhibition. In our original analysis, 100 μM PTX completely blocked CN synaptic potentials in only 3 out of 5 cells. However, upon re-analysis of all cells we found that the 2 cells with a seemingly suboptimal inhibition had irregular oscillations with high noise that distorted the outcome and conclusions. Taking this into consideration, careful re-inspection of individual traces revealed that inhibitory synaptic responses were in fact absent for all 5 out of 5 cells. We thank the Reviewer for pointing us to this.

f) Figures 7 and 8 are difficult to understand and need to be clarified.

Figures 7 and 8 have been overhauled and are now better explained (see new Figure plots and legends). Moreover, we have added one additional figure to explain the complex interaction phase and spiking in the population (Figure 9).

g) The health criteria for cells and the quality of recordings need further explanation.h) The authors also need to present the criteria for rejecting recordings on the basis of health or the value of series resistance compensation. The authors should explain whether there was a correlation between average membrane potential and the magnitude of intrinsic oscillations for the relevant cells.

We have now added the criteria to the Methods sections. Our criteria to reject cells were based on stability and access resistance value where cells typically showed access resistance values of less than 20 MΩ (on average 14.09 ± 8.74 MΩ (n=93) with 81% of cells ≤ 20 MΩ; 17% > 20 MΩ and < 40 MΩ; 2% > 40 MΩ and < 60 MΩ). If the recordings showed an access resistance of > 40 MΩ and/or the baseline was unstable for > 5 minutes the recording was rejected.

With respect to the correlation between average membrane potential and the magnitude of intrinsic oscillations, the distribution was virtually identical to what is observed in the model (see Author response image 1). Notably, there is an entire upper triangular part of the scatter that has no data points. This feature of the distribution of data points is explainable by the non-linear effect of Calcium low threshold (CaT) conductance on mean membrane potential, the upper triangle is void of data points, because oscillations amplitude is a function of CaT, which impacts mean membrane potential. Please find the new supplementary figure for model cells, copied below for convenience.

The Pearson R is also very close (0.2224 for experiment vs 0.22798 for the model). Though the p-value for the experiment was over 0.05 (experimental p-value = 0.066), there appears to be a trend. Both STO and mean membrane potential were measured during spontaneous activity.

**Author response image 1. sa2fig1:** 

i). Related to the previous comment, the authors should report in the Methods whether membrane potentials reported are corrected for the electrode junction potential, as well as the value of that potential.

Membrane potentials were not corrected for the electrode junction potential, the value of which was close to -7 mV. We have added this information to the Methods sections.

2) A discussion of how the results compare to natural firing patterns in vivo and natural physiological conditions would likely increase the impact of the paper.

We agree with the Reviewer and have added a few phrases in the Functional implications part of the Discussion on this topic.

3) Other text revisions are suggested to improve the clarity of the presentation of results.

The Results are now presented in a clearer fashion. In particular we have invested considerable effort in improving the comparisons of the modeling and experimental data.

Please refer to the individual reviews below for further details about essential revisions.Reviewer #1 (Recommendations for the authors):My main suggestion is to remove figures 7 and 8, which are difficult to follow (Figure 8) and somewhat not in line with the experimental results. In figure 7C an interval of 50ms seems to be extremely efficient in activating the neuron and resetting the oscillations, which is not in line with the main experimental (Figure 5D) or theoretical (Figure 4) results. Figure 8 is difficult to follow and its main input to the study is not clear. Is it the change in the number of affected neurons as a function of the interval?

Though this is not evident in the current version of the manuscript, this entire study originated with the observation that modeling predicts the phase responses from both types of input and thus predicted their interaction and consequences to network dynamics. The network model extrapolates from the individual cell data and produces predictions about the behavior of the network in a functional setting (see e.g., new Supplement figure –Figure 8—figure supplement 2- with network responses under perturbation). The ways in which the model is in line with the experimental data and which predictions it makes is now better explained in the Results and Discussion.

The main purpose of the model is to extend our understanding of the interaction between synaptic arborizations in the olive with the ongoing subthreshold oscillation in a network setting. We had also keenly noticed the 50 ms difference. We find that this difference is mostly dependent on properties of the GABA response – we had originally used exclusively a dendritic GABA, whereas now we use a combination of dendritic and somatic GABA, which leads to better alignment with the experimental data, and we include an enduring entrapment mechanism of GABA released within the glomeruli. Now the maximum rebound happens at one STO cycle (in the model ~100 ms), which we indeed had originally expected.

Our newly presented modeling data now better highlights the experimental findings and we thus feel the model contributes in substantial ways to the manuscript. Furthermore, it is fully open-source and in an open repository, which allows anyone to reproduce and extend our results. To improve the integration with the main text, and in line with the general advice of the Editorial Report (see above) we have extensively modified Figures 7 and 8, added a new Figure 9 regarding the expected responses. We have also added substantial supplements detailing (1) model tuning, (2) responses to stochastic input, (3) and phase differences.

Reviewer #2 (Recommendations for the authors):This study combines the use of transgenic mice, optogenetic approaches, and patch-clamp recordings in slices to examine the interplay between specific excitatory and inhibitory synaptic inputs and intrinsic oscillations in neurons of the inferior olive. Through analyses of the response phase and amplitude, the authors show convincingly that excitation has an especially powerful effect on response amplitude and firing when it occurs during post-inhibitory rebound. These findings build on many previous studies of this circuitry and are perhaps not surprising from a mechanistic point of view. However, the value of the work is derived from both the specificity achieved in selectively activating known sources of inhibition and excitation and also the thorough and systematic analyses the authors employ to examine the complex interactions between synaptic inputs and intrinsic voltage-gated conductances. Overall the study could be better supported in places, as documented below but provides a significant advance in understanding the circuitry underlying sensorimotor coordination.1. The presentation of health criteria for cells and the quality of recordings needs improvement. Membrane potentials are reported for both oscillating and non-oscillating cells, including an average and a range. This is a straightforward measurement for non-oscillating cells but not for oscillators. Presumably, the authors are simply averaging out the oscillations over a long time frame but this should be spelled out explicitly. Resting potentials of -40 mV would seem to be quite depolarized for a healthy neuron for either category and the variation in Vm across cells (nearly 27 mV) seems enormous.

For panel D of Figure 1 membrane voltage, Vm, was determined as the average of the voltage baselines preceding the current steps used for the I-V curve. When measuring the spontaneous spiking as a function of Vm, membrane voltage was taken as the average of a spontaneous activity trace (231 ± 153 s, ranging from 22 to 599 s).

Rn was measured as the slope of the linear part of the I-V curve. For the I-V curve, current steps were typically provided from -0.4 nA to 1nA. The linear part of the curve was usually between -0.4 nA and 0.1 nA.

In our observations, 27mV of STO amplitude is rare, but not unreasonable with respect to analysis of conductance spaces in single cell models. For instance, bifurcations from resting to oscillatory behavior via CaT in single cell modelling show that they are still in the oscillatory domain (see panel row 2, column 3 of our modelling figure above, which is now also a new supplementary dataset). Interestingly, such cells would be spontaneously spiking in the absence of gap junctions (supplementary cells parameter space A). For the two cells in our dataset with very depolarized membrane potentials (~ -40mV), we note that Vm was not corrected for the liquid junction potential. This applies to all cells reported. If corrected, Vm would be ~7mV more negative. We have now added this information to the main text.

2. The authors need to present the criteria for rejecting recordings on the basis of health, or the value of series resistance compensation, both of which are unaddressed currently. It would also be important to report whether there was a correlation between average membrane potential and the magnitude of intrinsic oscillations for the relevant cells. It would be expected that the average Vm of cells would strongly affect the relative balance between excitatory and inhibitory currents as well as their interactions with voltage-gated ion channels.

We agree with the Reviewer. Please see our answers highlighted above at the general points 1a, g and h.

3. Related to the previous comment, the authors should report in the Methods whether membrane potentials reported are corrected for the electrode junction potential, as well as the value of that potential.

Please see our answer to Essential revision point 1i.

4. It would be helpful in the Discussion if there was more attention paid to the natural firing patterns of inputs, and what physiological conditions might produce different time separations between excitatory and inhibitory inputs.

See Essential revisions point 2; we have added a few additional phrases to the general functional implications part in the Discussion on this topic.

Reviewer #3 (Recommendations for the authors):The oscillations in these cells are known to be highly voltage-dependent due to T-type Ca channels and h channels. Therefore, there is no surprise for the effects of inhibitory and excitatory inputs in altering the STOs. The main novelty of the findings in this paper involves the quantification of how the inputs affect the IO neurons. That said, there are numerous questions that the authors should address, some minor and some more significant.Specific suggestions:1. How were Vm and Rn measured? No details are given. Because both CaT and h channels are active at rest, how they were measured and then compared among cells is very important.

For figure 1 panel D, membrane voltage, Vm, was determined as the average of the voltage baselines preceding the current steps used for the I-V curve. When measuring spontaneous spiking as a function of Vm, membrane voltage was taken as the average of a spontaneous activity trace (231 ± 153 s, ranging from 22 to 599 s).

Rn was measured as the slope of the linear part of the I-V curve. For the I-V curve, current steps were typically given from -0.4 to 1 nA, the linear part of which was usually seen between 0.4 nA and 0.1 nA.

This information has now been added to the main text.

2. Although presumably CN and MDJ inputs are monosynaptic, some mention of this and whether there are any connections among IO neurons other than electrical would be helpful for readers outside of the field.

This information has now been added to the main text, while citing the appropriate Refs; the monosynaptic comment has been added to the introduction, while the scarcity of local interneurons has been mentioned in the discussion.

3. The rebound from a hyperpolarization is highly voltage-dependent. You can't just measure the amplitude from the baseline without knowing the voltage at the baseline and the voltage of the prior hyperpolarization. The rebound should be plotted as a function of the voltage of the prior hyperpolarization.

We thank the Reviewer for bringing this up. Please see below the plots in which baseline rebound amplitude is plotted either as a function of the prior hyperpolarization or pre-stimulus membrane voltage. This information has now been added to the Results section (in Supp Figure).

4. In figures 2C and 3C there are no scale bars for either x or y axes.

This has been corrected.

5. In Suppl Figure 1, were the synaptic blockers added sequentially and thus additive, or are the responses shown from only a single blocker?

As highlighted above at the general points, this information has now been added to the legends of Suppl. Figure 1 (now Figure 3—figure supplement 1). The drugs were added sequentially: first PTX, then CNQX, and finally, if possible, also APV. PTX and CNQX were added in all cells (8 in total), whereas APV could only be successfully added in 4 out of 8 cells, due to the relatively long duration of the required recordings (>80 mins).

6. No figure was provided for the effect of PTX on CN inputs. This should be included. Also, why didn't 100uM PTX fully block the inhibition? Some ideas for this would be useful.

This information has now been added. As explained in detail above at the general point (e), we now show a full blockage of inhibition following application of 100 μm PTX (see also panel below, which has been included as a Figure 2—figure supplement 1) We thank the Reviewer for drawing our attention to this point.

[Editors' note: further revisions were suggested prior to acceptance, as described below.]

The manuscript has been improved but there are some remaining issues that need to be addressed, as outlined below:Reviewer #3 (Recommendations for the authors):I have even less confidence in the rigor of the electrophysiology than I did in the previous version. I asked:1. For details on Vm and Rn. The response for Vm was ok, but not Rn. Where is the linear portion of the I-V curve (actually it should be called a V-I) curve? No V-I is shown so I have no idea whether there was a linear portion and what the rest of the V-I plots look like.

Please note that all the curves are uploaded to the DRYAD repository (folder: “Figure 1”, filename: Rn calculations) along with the Manuscript. As an example, please see the plot in Author response image 2. The red rectangle highlights the data points selected for the linear regression to determine input resistance. We note that (1) linearity is not completely absent beyond the range selected for Rn calculations (in this case linearity is observed up to 0.4 nA) and that (2) even if we had used all the data points from the curve, it wouldn't have significantly altered the computed input resistance. For instance, if all data points in the example shown would have been included for the linear regression analysis, the input resistance and R^2^ would have been 26.61 ΩM and 0.90, instead of 27.93 ΩM and 0.99. This was true for all cells (see DRYAD).

2. Details about rebound being v. dependent. They now show plots, and indeed rebound varied 2-4 fold depending on Vm, but as far as I can tell, none of this was used in their analyses such as in Figure 2 and elsewhere.

While it's true that rebound amplitude exhibits a voltage-dependent nature, we must emphasize that variations in membrane voltages across different cells – differences that merely fall within a range of 2 to 3 mV – only have a negligible impact in this particular instance. Such minute differences are too small to significantly alter our observations. Most critically, our assertions are firmly rooted not just in empirical data, but also have been rigorously affirmed by computational modeling studies that corroborate our experimental findings.

3. I asked about 100uM PTX. They now show data from what appears to be one cell but claim that "reanalysis" indicated a complete block in 5 of 5. Where are the data?

The reanalyzed data may seem to originate from a single cell, but they actually average the results from all five cases, as every cell tested showed identical behavior. Specifically, PTX entirely eliminated GABAergic responses triggered by CN stimulation across the board. All data, including those of the two reanalyzed cells (17n16062, due to irregular oscillations in half the traces, and 19409047, due to noise in two traces), have been uploaded to DRYAD under the path: Folder Figure 2> Pharmacology>ABF files.